# Phenotypic, Physiological, and Molecular Response of *Loropetalum chinense* var. *rubrum* under Different Light Quality Treatments Based on Leaf Color Changes

**DOI:** 10.3390/plants12112169

**Published:** 2023-05-30

**Authors:** Yifan Zhang, Yang Liu, Lin Ling, Wenwen Huo, Yang Li, Lu Xu, Lili Xiang, Yujie Yang, Xingyao Xiong, Donglin Zhang, Xiaoying Yu, Yanlin Li

**Affiliations:** 1College of Horticulture, Hunan Agricultural University, Changsha 410128, China; zyfan@stu.hunau.edu.cn (Y.Z.); 418272846@stu.hunau.edu.cn (Y.L.); huowenwen0726@stu.hunau.edu.cn (W.H.); 1304990057@stu.hunau.edu.cn (Y.L.); xiongxingyao@caas.cn (X.X.); 2Engineering Research Center for Horticultural Crop Germplasm Creation and New Variety Breeding, Ministry of Education, Changsha 410128, China; 3School of Economics, Hunan Agricultural University, Changsha 410128, China; lljjxy@hunau.edu.cn; 4Kunpeng Institute of Modern Agriculture, Foshan 528225, China; 5Agricultural Genomics Institute at Shenzheng, Chinese Academy of Agricultural Sciences, Shenzhen 518120, China; 6Department of Horticulture, University of Georgia, Athens, GA 30602, USA; 7Hunan Mid-Subtropical Quality Plant Breeding and Utilization Engineering Technology Research Center, Changsha 410128, China; 8School of Biological Sciences, Nanyang Technological University, 60 Nanyang Drive, Singapore 637551, Singapore

**Keywords:** *Loropetalum chinense* var. *rubrum*, light quality, leaf color, gene expression, anthocyanin

## Abstract

Light quality is a vital environmental signal used to trigger growth and to develop structural differentiation in plants, and it influences morphological, physiological, and biochemical metabolites. In previous studies, different light qualities were found to regulate the synthesis of anthocyanin. However, the mechanism of the synthesis and accumulation of anthocyanins in leaves in response to light quality remains unclear. In this study, the *Loropetalum chinense* var. *rubrum* “Xiangnong Fendai” plant was treated with white light (WL), blue light (BL), ultraviolet-A light (UL), and blue light plus ultraviolet-A light (BL + UL), respectively. Under BL, the leaves were described as increasing in redness from “olive green” to “reddish-brown”. The chlorophyll, carotenoid, anthocyanin, and total flavonoid content were significantly higher at 7 d than at 0 d. In addition, BL treatment also significantly increased the accumulation of soluble sugar and soluble protein. In contrast to BL, ultraviolet-A light increased the malondialdehyde (MDA) content and the activities of three antioxidant enzymes in the leaves, including catalase (CAT), peroxidase (POD), and superoxide dismutase (SOD), in varying degrees over time. Moreover, we also found that the *CRY*-like gene, *HY5*-like gene, *BBX*-like gene, *MYB*-like gene, *CHS*-like gene, *DFR*-like gene, *ANS*-like gene, and *UFGT*-like gene were significantly upregulated. Furthermore, the *SOD*-like, *POD*-like, and *CAT*-like gene expressions related to antioxidase synthesis were found under ultraviolet-A light conditions. In summary, BL is more conducive to reddening the leaves of “Xiangnong Fendai” and will not lead to excessive photooxidation. This provides an effective ecological strategy for light-induced leaf-color changes, thereby promoting the ornamental and economic value of *L. chinense* var. *rubrum*.

## 1. Introduction

Ornamental plants are vital components of the landscape and the ecological systems of urban and countryside environments [1]. *Loropetalum chinense* var. *rubrum*, a colored-leaved tree, is a woody plant with characteristic vividly colorful leaves and flowers that is easily shaped and trimmed, has high adaptability, and is native to tropical and subtropical regions [2,3]. As it contains anthocyanins and other bioactive components, the plant can be used for medicine, e.g., the leaves can be used for hemostasis, the roots and leaves can be used for bruises and can be effective in removing blood stasis. Therefore, *L. chinense* var. *rubrum* has become an important plant for gardening [4], and has greatly promoted local economic development [5]. However, the composition and contents of the anthocyanins in the leaves are significantly different to the original *L. chinense*. The leaf color of *L. chinense* var. *rubrum* is affected by many environmental factors, including temperature, water supply, and light intensity or light spectrum [6,7]. Unsuitable cultivation conditions and growth environments result in the reduction and degradation of the anthocyanins in the leaves, which seriously affects their color, reduces their ornamental value and commodity traits, and causes substantial economic loss. Varying leaf color and vulnerability to environmental stresses reduce the market value of this plant. Hence, it is desirable to develop a simple and convenient method of controlling the leaf color in order to improve the market value of the plant. The method of improving plant color by varying light quality is widely used because of its low cost and easy operation.

Light is an indispensable factor in plant growth and development. Light quality means the color or wavelength reaching the surface, which strongly affects the physiological, morphological, and biochemical parameters [8,9,10,11]. For example, previous literature reported that blue light (BL) and UV-A were more conducive to the accumulation of anthocyanins in plants [12,13,14,15]. Similarly, the increase in soluble sugar and soluble protein under BL and UV-A also leads to the accumulation of anthocyanins. They provide the energy and material basis for anthocyanin synthesis [15]. The soluble sugar content under a BL treatment was the highest, which effectively promoted the carbon metabolism of *Liquidambar formosana* leaves, thus providing raw materials for the large synthesis of anthocyanins, thus promoting the redness of leaves [16]. In addition, BL and UV-A may cause changes in the leaf color of plants by stimulating anthocyanin synthesis [17].

Light of a specific wavelength may cause the accumulation of unwanted and harmful reactive oxygen species (ROS) in plant leaves, also affecting the metabolism of anthocyanins. UV-A may increase the accumulation of anthocyanins, while it also induces ROS production in large quantities, including superoxide anion (O_2_^−^), hydrogen peroxide (H_2_O_2_), and hydroxyl radical (-OH) [18]. Superoxide dismutase (SOD) is an essential antioxidant enzyme and a potent scavenger of ROS [19]. It catalyzes the superoxide anion radical disproportionation producing oxygen and hydrogen peroxide. Oxidation and antioxidant balance play crucial roles in plants [20]. The cooperation of catalase (CAT) and glutathione peroxidase (GSH-Px) can decompose H_2_O_2_ into O_2_ and H_2_O. CAT and GSH-Px also synergized with SOD to eliminate oxidative damage. Malondialdehyde (MDA) is one of the products of membrane lipid peroxidation, which indirectly reflects ROS damage to cells and organisms. Moreover, it was also used as one of the detection indexes of cell senescence when facing and enduring damage [21]. Antioxidant enzymes and anthocyanins play a crucial role in plant antioxidant processes, and they have a complex relationship. Of these, it has been reported that anthocyanin content and antioxidant enzyme activity were proportionally increased after light treatment in *Quercus* [22]. In addition to a phenotypic and physiological responses, plants have complex molecular mechanisms of response to different light qualities. Photoreceptors are cryptochromes (CRY) that respond to BL and UV-A (in the wavelength range 320–500 nm). CRY may activate specific downstream light signal transduction pathways with the absorption of specific light [23]. It was found that the *Solanum lycopersicum* with overexpression of CRY1a accumulated anthocyanins when it was exposed to BL [24]. Elongated hypocotyl 5 (HY5) is a fundamental structural domain/Leu zip transcription factor that directly targets constitutively photomorphogenic 1 (COP1) and acts as a central positive regulator of light signaling. *SIHY5* silencing and *CRY1a* overexpression led to a significant decrease in anthocyanin accumulation in *Arabidopsis thaliana* (L.) Heynh and other plants. [25,26,27]. CRY1/CRY2-COP1 interactions triggered by BL created conditions of HY5 and MYB1 (v-myb avian myeloblastosis viral oncogene homolog) binding to downstream anthocyanin structural (AS) genes, further inducing anthocyanin accumulation in *Solanum melongena* [28,29]. HY5 can act as an accelerator for direct binding to the promoters of light-inducible genes, including structural and regulatory genes, to increase their abundance for the accumulation of anthocyanins [29]. Moreover, a member of the B-box zinc finger gene (BBX) family is a target of HY5. It is involved in the regulation of anthocyanin accumulation [30,31,32,33].

Under the control of light quality, light receptors receive light signals and transmit them to downstream genes, which together regulate transcription factors, resulting in the synthesis of a series of downstream anthocyanin structural (AS) genes [34]. The anthocyanin glucoside metabolic pathway is one of the most-studied biological metabolic pathways [35]. Li et al. [36] found that BL can increase the expression of the structural genes chalcone synthase (CHS) (the first key enzyme for the plant synthesis of flavonoids) and dihydroflavonol 4-reductase (DFR) (a key enzyme in the flavonoid biosynthesis pathway) in anthocyanin synthesis in *Chrysanthemum* (*Chrysanthemum* × *morifolium* Ramat.). There are also many results showing that the expression differences of most AS genes, such as CHS, DFR, anthocyanidin synthase (ANS) (a key enzyme at the end of the plant anthocyanin biosynthesis pathway, catalyzing the conversion of colorless anthocyanins into colored anthocyanins), and UDP-glucose: flavonoid 3-O-glucosyltransferase (UFGT) (the last enzyme in the process of anthocyanin synthesis, which can catalyze unstable anthocyanins into anthocyanins), depend on the expression of transcription factors (such as MYB) [37,38]. A number of studies have shown that the expression of MYB can be activated under BL and UV light in apples [39,40,41], purple pak choi [42], tomato [43,44], peach [45], and other peels to varying degrees, thereby regulating the upregulation of anthocyanin synthesis-related genes, and ultimately promoting red-colored peel results. Therefore, based on molecular data, using BL and UV-A radiation to increase anthocyanin production is feasible. Overall, further in-depth studies on the relationship between anthocyanin synthesis and the physiological and molecular response mechanism to light qualities are needed.

According to the above introduction, we hypothesized that the reason for the red leaf color of *L. chinense* var. *rubrum* under light quality treatment may be that light affects the expression of anthocyanin-related genes to promote their synthesis, and reduce the stress response of plants. In this study, physiological, biochemical, and molecular verification methods were used to explore the light quality that can best promote color improvement in *Loropetalum chinense* var. *rubrum* leaves without causing light stress, and to verify the relationship between some physical and chemical indicators and anthocyanin content under different light qualities. By analyzing the changes in pigment, soluble sugar and protein, antioxidant enzyme activity, and anthocyanin synthesis-related genes in leaves under BL and UL, the mechanism of color improvement in *L. chinense* var. *rubrum* leaves was examined. An effective *L. chinense* var. *rubrum* leaf color improvement scheme was proposed, which would improve its market competitiveness and provide a theoretical basis for artificially enriching the leaf color of *L. chinense* var. *rubrum*. Finally, in our research discussions, we hope to be able to translate this basic scientific knowledge into opportunities for agricultural production, and to provide ideas for future research directions for the subject.

## 2. Results

### 2.1. Changes in Leaf Color Phenotype under Different Light Quality Treatments

It can be seen from Figure 1 and Table 1 that the leaf color changed with time, and the leaf color descriptions at 0 d were olive green in every group. The Munsell color system (hue, value, and chroma; these will be abbreviated as H, V, C later.) of leaves at 0 d under WL and BL + UL were: 2.4 GY, 4.90, and 2.00; and 1.2 GY, 4.90, and 2.10, respectively, and the leaf color qualitative descriptions were all olive green. After seven days of treatment under WL and BL + UL, the values of H, V, and C were: 9.9 YR, 3.90, and 2.00; and 9.9 YR, 3.90, and 1.9, respectively, and the leaf color descriptions had all changed to yellowish-brown. The H, V, and C values of leaves under BL treatment at 7 d were: 0.53 YR, 3.50, and 1.80, and the leaf colors were qualitatively described as reddish-brown. The leaves treated with the above three light qualities could turn red, and the redness increased the most under BL treatment. Under UL treatment, the H, V, and C of leaves at 7 d were: 1.5 GY, 4.80, and 1.60, respectively. The qualitative description of the leaves became olive green, and the color of the leaves did not change in the red direction. 

Furthermore, by observing the distribution of pigments in leaf cross-sections by freehand sectioning, we found that anthocyanins were mainly concentrated in the palisade tissue (PT) and less in the upper epidermis (UE), lower epidermis (LE), and spongy tissue (ST), which largely determined the color we observed in the leaves. In the leaves under BL treatment, we could observe more clearly that the degree of redness in the palisade tissue was higher on 7 d than under WL, BL + UL, and UL treatments. The changes in pigmentation in the cross-sectional structure were largely consistent with the description of the changes in leaf color. 

### 2.2. Changes in Pigment Contents under Different Light Qualities with Time

It is well known that pigments and the proportion of pigments such as chlorophyll, anthocyanin, and carotenoid determine the color of leaves [46,47]. As shown in Figure 2, chlorophyll (a + b) content increased over time under different light qualities reaching a maximum at day 7, except for WL, where the increase was insignificant. The order of magnitude of increase was: UL (139.9%) > BL + UL (56.5%) > BL (32.4%). The carotenoid content decreased with time under WL treatment. The content decreased first, and then increased under BL treatment, reaching the maximum at 7 d. The carotenoid content increased with time under BL + UL and UL treatments, increasing the most under UL treatment at 7 d, 122.0% compared with 0 d. The anthocyanin content also showed an increasing trend with time. Under BL treatment, it reached the maximum (0.102 mg/g) at 7 d, increasing by 137.2% compared with 0 d, followed by BL + UL treatment (0.096 mg/g), while UL treatment resulted in the most negligible anthocyanin content increase (0.067 mg/g). The changing trend of total flavonoid content was consistent with that of anthocyanin content. With the increase in time, the anthocyanin content was significantly higher at 7 d than at 0 d, and the increasing order was: BL (153.4%) > BL + UL (108.1%) > WL (87.3%) > UL (76.6%). The contents of the above four pigments all increased most significantly under BL treatment.

### 2.3. Changes in Soluble Sugar and Soluble Protein Content with Time under Different Light Qualities

As can be seen from Figure 3, the soluble sugar content and soluble protein content increased gradually over time under all light qualities, with the soluble sugar content increasing significantly to the highest level at 7 d under BL treatment, increasing by 1087.35% compared with 0 d, followed by BL + UL treatment, increasing to 778.54% at 7 d. Compared with 0 d, the content increased by 538.89% under WL treatment, while under UL treatment, it increased the least, increasing by only 197.93% at 7 d compared with 0 d. The soluble protein content peaked at 3 d under UL treatment with an increase of 28.72% over 0 d. At 7 d, the content under BL treatment reached its peak, increasing by 86.57% compared with 0 d, followed by the content under BL + UL treatment, with an increase of 74.33% over 0 d, while WL treatment only increased the content by 24.77% over 0 d. In the leaves of “Xiangnong Fendai”, the BL treatment was more beneficial to a significant accumulation of soluble sugars, while treatment with UL was more beneficial for a significant accumulation of soluble proteins, followed by BL + UL treatment. 

### 2.4. Stress Injury and Self-Repair of the Antioxidant System during Leaf Growth under Different Light Qualities

It can be seen from Figure 4 that, under UL, the activities of antioxidant enzymes CAT, SOD and POD were generally higher compared to other light treatments. UL treatment resulted in the maximum value of SOD (454.7 U/g) at 7 d, which was significantly higher than that at 0 d (173.1%). The activity of SOD in leaves peaked at day one of exposure to WL (374.5 U/g), BL + UL (463.5 U/g), and BL (331.0 U/g), and then decreased. POD enzyme activity under UL treatment jumped to the maximum (951.1 U/g) at 1 d, a significant 135.3% increase over 0 d, while under WL (832.2 U/g), BL + UL (927.3 U/g) and BL (998.6 U/g) treatments, it reached the maximum at day 5. CAT activity under UL treatment reached the maximum (725.5 U/g) at 5 d, which was significantly higher than that at 0 d (328.0%). The MDA content also showed a trend of increasing first and then decreasing in each treatment group. The MDA content of the plant leaves under UL treatment reached the maximum value (53.4 nmol/g) on the third day, which was significantly higher than that at 0 d (137.3%). The maximum content was attained on the fifth day under WL treatment (50.04 nmol/g). Therefore, the antioxidant enzyme activity and MDA content increased the most under UL treatment, followed by treatment with BL + UL.

### 2.5. Correlation between Physiological and Biochemical Indicators and Anthocyanin Content in the Leaves

Under WL treatment, the anthocyanin content in leaves was significantly positively correlated with CAT enzyme activity and soluble protein content (Figure 5). Under BL + UL treatment, there was a significant positive correlation between total flavonoid content and soluble protein content. The anthocyanin content of the leaves under BL treatment was significantly positively correlated with a* value (representing the redness of the leaves), soluble protein content, and soluble sugar content. No significant parameters related to anthocyanin synthesis were found under UL treatment. This result proves that the larger the a* value, the higher the anthocyanin content in the leaves. In most cases, in terms of the content of soluble sugar and soluble protein in leaves, the higher the content of both or one of them, the more conducive it is to the accumulation of anthocyanins.

### 2.6. The Gene Expression of Related Anthocyanin Synthesis under Different Light Qualities

We also analyzed the expression patterns of some light signaling and AS-related genes using qRT-PCR (Figure 6). According to our findings, *augustus 35127* and *augustus 55502*, two similar genes to *CRY* in ‘Xiangnong Fendai’, were both significantly increased to a maximum at different times under different treatments and then decreased. Under BL treatment, the relative expressions of *augustus 35127* and *augustus 55502* were 5.59- and 7.55-fold higher at 5 d and 3 d than at 0 d. *Augustus 55502* expression reached its maximum at 3 d under WL, BL and UL treatments, which was 5.99-fold, 7.54-fold and 19.61-fold higher than at 0 d, respectively. The *augustus 64086* and *augustus 18354* genes are the two analogs of *HY5* downstream of *CRY*. The *augustus 64086* gene reached its highest relative expression at 5 d under BL treatment, 312.10-fold higher than at 0 d. In contrast, *augustus 18354* reached its maximum at 1 d under BL + UL, BL, and UL treatments, all significantly higher than at 0 d. A *BBX*-like gene (*augustus 08157*), a target of *HY5*, reached its maximum at 1 d of WL, BL + UL, and BL treatments, significantly higher than at 0 d, which was 26.93-fold, 35.41-fold, and 60.96-fold higher than at 0 d, respectively. The relative expression of a similar transcription factor *MYB* (*augustus 33908*) was highest at 1 d of BL treatment, 4.00-fold higher than at 0 d.

The *augustus 35565* and *augustus 57090* genes, two *CHS*-like genes of the pre-anthocyanidin synthesis gene in *L. chinense* var. *rubrum*, varied in expression with the time of light quality treatment, reaching their highest at 5 d and 3 d of BL treatment, respectively, both significantly higher than when untreated at 0 d. The expression of these genes was 287.04 and 34.34 times higher than that at 0 d. The late anthocyanin synthesis genes included *augustus 47340* and *augustus 36047* (*DFR*-like genes), *augustus 44839* (*ANS*-like gene), and *augustus 48540* (*UFGT*-like *gene*), with *augustus 47340* and *augustus 36047* reaching maximum expression at 7 d under both BL + UL and BL treatments, being 5.36 and 4.56, and 5.36 and 3.91 times higher than at 0 d, respectively. The *augustus 44839* gene reached maximum expression at 5 d under BL treatment, being 1.46 times higher than at 0 d, while the expressions under WL and BL + UL treatments were significantly higher at 7 d than at 0 d, being 1.26 and 1.48 times higher than at 0 d. Expression of *augustus 48540* reached a maximum at 7 d under UL treatment, significant at 100.98 times higher than at 0 d, while expression under BL treatment reached a maximum at 5 d, 35.50 times higher than at 0 d. This shows that the relative expression of most genes was more significantly upregulated under BL treatment than under the other treatments.

### 2.7. The Gene Expression of Related Antioxidant Enzymes under Different Light Qualities

From Figure 7, we can see that there was no significant change in the expression of antioxidant enzyme-related genes at 1 d under WL treatment, while BL + UL, BL, and UL treatments all started to show significant changes in expression at 1 d. Both *augustus 35171* and *augustus 13933* (*SOD*-like gene) reached maximum expression at 7 d under BL treatment, which was significantly higher than that at 0 d. Both *augustus 35171* and *augustus 13933* (*SOD*-like gene) expression reached the maximum value at 7 d under BL treatment, which was significantly higher, at 58.32 and 14.70 times, respectively, than that at 0 d. The expression of *augustus 46250* (*POD*-like gene) reached the maximum value at 1 d under BL + UL treatment and then decreased, significantly higher than that at 0 d. The expression of *augustus 11979*(*POD*-like gene) peaked at 7 d, 72.76 times higher than that at 0 d, the most significant change compared with other treatment groups. The relative expression of the *CAT*-like gene was upregulated under UL treatment, with *augustus 68148* showing the highest significant increase at 5 d of UL treatment. The relative expression of the *CAT*-like gene was upregulated under UL treatment, with *augustus 68148* showing the most significant increase at 5 d of UL treatment, 284.65-fold higher than at 0 d, and *augustus 12226* showing a tremendous increase at 7 d of UL treatment, 158.47-fold higher than at 0 d, followed by BL + UL treatment and WL treatment, 115.74 and 33.05 times higher than at 0 d. Thus, the relative expression of antioxidant enzyme-related genes was significantly increased by UL treatment at different times. 

## 3. Discussion

### 3.1. Different Changes in Leaf Phenotype with Time under Different Light Qualities

Increasingly, more flowers and more colorful foliage plants have gradually entered the research fields of scholars. Leaves are able to be viewed for longer than flowers, so it is more meaningful to study the regulation of leaf color. To accurately define and describe leaf color, the quantitative analysis of leaf color phenotype is required. We transformed a colorimeter’s L*, a*, and b* values into H, V, and C values. Using CIELab to describe the color, it was found that the leaf color under BL treatment changed from olive green to reddish-brown, and the color of the leaves clearly turned red. The redness of the leaves under WL and BL + UL treatments was lighter, while the leaves under UL treatment turned greener than the former. The quantitative analysis results are consistent with the observation results. This leaf color quantitative analysis method is more objective and fact-based than the traditional description.

In the cross-section structure, we could see that anthocyanins were mainly distributed in the upper epidermis and palisade tissue. Whether this was because, in the light quality setting of our experiment, the direction of light was from top to bottom, resulting in the anthocyanin accumulation to be mainly distributed in the upper part of the leaf, we have not yet proved.

### 3.2. Changes in the Physicochemical Properties of Leaves Induced by Different Light Quality Treatments with Different Times

By combining the determination of leaf pigment content, we found that, under BL treatment, the anthocyanin content and total flavonoid content in the leaves increased significantly compared with the other treatment groups, followed by BL + UL and WL treatments. Meanwhile, under UL treatment, the chlorophyll and carotenoid content increased significantly. The soluble sugar content of the osmotic adjustment substance was the highest at 7 d under BL treatment. In contrast, soluble protein peaked at 3 d under UL treatment, and then decreased with time, but compared with the other treatment groups, the content at 7 d increased the most compared with 0 d. Therefore, in our study, it could be seen that BL treatment was more conducive to the accumulation of anthocyanins, which increased the redness of the leaves. In contrast, BL + UL treatment was not as effective as BL, because it contained UL that is not conducive to an increase in anthocyanins. The WL of the control group was not as effective as BL + UL, because it was a full spectrum of light, including BL, but containing a small proportion. Previous studies have shown that BL can promote the accumulation of anthocyanins in plants and make plants redder [48,49,50], which is consistent with our conclusions.

From the results, we also found that the leaf phenotype under WL treatment at 5 d was significantly redder than that at 0 d, but the anthocyanin content did not increase significantly. From previous studies conducted by our group [51], 185 flavonoids and 22 polyphenols other than flavonoids were identified in *L. chinense* var. *rubrum* (including 19 phenolic acids and 3 proanthocyanidins). Non-anthocyanin polyphenols such as flavonoid glycosides and phenolic acids are auxiliary pigment components that affect the leaf color of *L. chinense* var. *rubrum*. The reason for this phenomenon may be due to the increase in the content of some auxiliary pigments in the leaves. It can also be seen from our total flavonoid content data that the total flavonoid content was significantly increased at 5 d compared with 0 d. However, this is only one of our speculations and has not been verified, so we will try to substantiate this statement in future experiments.

Nevertheless, many studies have found that UV-A is conducive to promoting the accumulation of anthocyanins in plants [52]; this is contrary to our experimental results. This phenomenon may be due to the additional damage caused by UV-A with different light intensities. The ultraviolet light intensity used in this study was relatively high, which may stress plants. Mao et al. [53] found that low light intensity UV-A (100 μmol·m^−2^·s^−1^) was more conducive to increased soluble sugar and soluble protein content in cabbage leaves. In comparison, high light intensity UV-A (400–1000 μmol·m^−2^·s^−1^) was only conducive to an increase in soluble protein content, which is also consistent with the results of this study. Meanwhile, our study also found that different light qualities and different times could also change the antioxidant enzyme activity in leaves. In the early stage of UL treatment, MDA in leaves increased, and the three antioxidants increased more obviously than in other treatment groups at different times, followed by BL + UL treatment. This indicated, from the side, that UV-A produced light stress, and the three antioxidant enzyme activities increased successively to offset the oxidation produced by stress, to finally reduce MDA.

### 3.3. Anthocyanin Content under Different Light Quality Conditions Has a Different Correlation with Other Physicochemical Indexes

Based on our analysis of the correlation between each index and anthocyanin content accumulation under different light qualities, the results showed that there was no significant correlation between anthocyanin content and physicochemical parameters under UL treatment. This may also explain why the accumulation of anthocyanin content under UL treatment is lower than in other groups. The reason for this may be that the ultraviolet light used in our experiment caused the normal physiological state of plants to be disturbed, the self-defense system to be affected, and the ability to synthesize various nutrients to decrease in varying degrees. The anthocyanin content under BL and BL + UL treatments, with blue light participation, was significantly positively correlated with multiple indicators, which may stimulate plants to produce more nutrients under BL treatment or actively regulate the synthesis pathway of nutrients to protect themselves.

### 3.4. Anthocyanin Synthesis-Related Genes and Antioxidant Enzyme Synthesis-Related Genes Respond to the Expression Regulation of Different Light Qualities

In recent years, molecular biology and genetic studies have identified several components involved in the light-stimulated anthocyanin synthesis response, including photoreceptors and AS structural genes. In petunia, MYB regulators and AS structural genes were also identified as being involved in anthocyanin synthesis under light conditions [54,55]. In order to elucidate the molecular response mechanism of *L. chinense* var. *rubrum* based on leaf color changes under different light quality treatments, we used qRT-PCR technology to analyze the expression of related vital genes. Our research results found that *CRY*-like genes were upregulated after receiving BL signals and activated downstream *HY5*-like genes and *BBX*-like genes. BBX proteins play a role with HY5 at the translation or transcription level [56]. Different light qualities may lead to the activation of specific BBX-HY5 pathways, which in turn cause the transcriptional regulation of *MYB*-like genes, the activation of downstream *AS*-like genes, and ultimately the accumulation of anthocyanins, which confirms previous studies [57]. However, under UL, the changes in these genes were not noticeable compared with BL. In the determination of antioxidant enzyme-like genes, it was found that UL treatment induced the upregulation of more antioxidant enzyme-like genes, which also indicated that UL in this study caused stress to the plants, thus inducing a large amount of ROS production, leading to an increase in MDA content. Finally, antioxidant measures were initiated in the leaves under UL treatment, thereby significantly improving the activity of antioxidant enzymes to eliminate the damage caused by ROS. The reason for the lower oxygen stress under BL treatment may be due to the increase in anthocyanin content in the leaves under BL treatment, which inhibits the production of ROS. Anthocyanins can resist UV radiation, improve UV-barrier function, protect plant DNA from damage, and maintain cell differentiation and other everyday life processes [58].

We used a diagram to illustrate the response of the plants after receiving blue and ultraviolet light signals (Figure 8). It can be seen that blue and ultraviolet light have some of the same photoreceptors, but can cause the same or different responses in plants. Based on the physiological effects of blue and ultraviolet light on the different stages of plant development, for example, it can affect dry mass, flowering time, branching, metabolism, etc. Blue and UV radiation can be more widely used in crop production. Interestingly, CRY1 senses UV-A rather than BL and promotes anthocyanin synthesis via HY5. Similar to UVR8, CRY1 is also involved in the expression of the genes related to photooxidative-stress tolerance in the signaling pathways under UV-A and BL.

## 4. Materials and Methods

### 4.1. Plant Materials

Mature (two-year-old) *L. chinense* var. *rubrum* “Xiangnong Fendai” trees (with leaf-coloring of light-sensitive material) (National Forestry and Grassland Administration of China, certificate number of 20220206) growing in an ornamental horticultural experiment field in Hunan Mid-Subtropical Quality Plant Breeding and Utilization Engineering Technology Research Center, Hunan Agricultural University, Changsha, Hunan, China (28°12′ N, 112°59′ E) were chosen for the study. Water and fertilizer management that maintained the same ratio of cultivation substrate was used for all experimental materials, with the following soil conditions: garden soil: vermiculite: perlite = 3:1:1.

### 4.2. Light Quality Settings

Based on previous studies, light sources more favorable for anthocyanin synthesis were utilized as our experimental light sources, so as to screen for light sources more favorable for anthocyanin synthesis in L. chinense var. rubrum. In order to remove other influencing factors, the following experiments were carried out in the artificial climate control room with custom-made LED lamps: white light (WL) (plant growth spectrum), WL as control, blue light plus UV-A (BL + UL) (460 nm + 320 nm) (light ratio 1:1), blue light (BL) (460 ± 5 nm), and UV-A (UL) (320–400 nm). The light intensity was controlled by a light intensity meter at approximately 200 μmol/m^2^/s. The artificial climate control room had a light cycle precisely regulated by an automatic control switch (day/night: 14 h/10 h), a temperature of 24 °C/20 °C (day/night), and a humidity of 65–75%.

### 4.3. Leaf Observation

The microscopic observation of the pigment distribution in the transverse structure of the leaf was carried out as follows. The second and third young leaves in a branch, counted from the top of the branch (approximately 5–6 leaves), were selected for observation under different treatments and at different time points (the selection criteria for the following test leaves were the same to ensure the consistency of the conditions). Distilled water was added to the middle of a clean slide and potatoes were cut into strips to be used as fixed objects, with a small opening at the front of the strip to hold the leaf in the middle. After cutting several strips with a razor blade, a thin enough cross-section of the leaf was selected, covered with a coverslip, and the distribution of pigments in the cross-section structure was observed under an optical microscope (Leica D-35578, Wetzlar, Germany) with the photographs saved.

The leaf color was measured at 0, 1, 3, 5, and 7 d after different light treatments, and the choice of leaf material was the same as above. All the measured values of L*, a*, and b* were quantitatively derived using the YS 3020 spectrophotometer (3 nh, China). With three leaves per measurement, the lightness (L*), redness (a*), and yellowness (b*) measured by the colorimeter was converted into hue (hue, H), lightness (value, V), and saturation (chroma, C) of the Munsell color system. The qualitative description was obtained by comparing the H, V, and C values with the ISCC-NBS color name representation [59,60]. 

### 4.4. Determination of Pigment Content

(1) Determination of Chlorophyll Content

At 0, 1, 3, 5, and 7 d after light quality treatment, three randomly selected leaves of the same node from each group were used as samples, with the choice of leaf material the same as that in Section 4.3. The chlorophyll and carotenoid content was determined by direct ethanol extraction. The leaves were treated with 95% ethanol for 15 mL at 4 °C for 24 h in the dark until the leaf strips turned white. The supernatant was aspirated, and the absorbance values were measured at 470, 645, and 663 nm (using A470, A645, and A663 representations, respectively; V was the volume of the extract; W was the mass of leaf sample) [61].
Chlorophyll a (Chl a) = 12.7 × A663–2.69 × A645 × V/1000 W (mg/g)(1)
Chlorophyll b (Chl b) = 22.9 × A645–4.68 × A663 × V/1000 W (mg/g)(2)
Carotenoids (Car) = (1000 × A470–3.27 × Chl a-104 × Chl b)/229 × V/1000 W (mg/g)(3)
Chl (a + b) = Chl a + Chl b (mg/g)(4)

(2) Determination of anthocyanin content

Samples were taken at 0, 1, 3, 5, and 7 d for the determination of anthocyanins, and the choice of leaf material was the same as that in Section 4.3. The total anthocyanin content of the leaves was determined using the pH difference method [51,62,63]. Approximately 3 g of the sample was quickly ground into a fine powder in liquid nitrogen, and 20 mL of the extract was added (0.05% methanolic hydrochloric acid solution), extracted for 24 h at 4 °C, and centrifuged for 20 min at 8000× *g*. The supernatant was transferred to a clean test tube, and buffer A (0.4 mol/L KCl, pH adjusted to 1.0 with HCL) and buffer B (0.4 mol/L citric acid, pH adjusted to 4.5 pH with NaH_2_PO_4_) were added to each 1 mL of supernatant. The absorbance at 510 nm and 700 nm was measured for solution A and solution B, respectively. Measurements were repeated three times for each treatment.

### 4.5. Determination of Total Flavonoids, Soluble Sugar, and Soluble Protein Content

The contents of total flavonoids, soluble sugar, and soluble protein in the leaves were determined using kits (total flavonoids assay kit: ZCSO873, Shanghai zcibio technology Co., Ltd., Shanghai, China; plant soluble sugar assay kit: BC0030, Solar-bio, Beijing, China; plant protein quantification assay kit: A045-2, Jiancheng, Nanjing, China), and the choice of leaf material was the same as that in Section 4.3.

### 4.6. Determination of MDA Content and Antioxidant Enzyme Activity 

We investigated the malondialdehyde (MDA) content and the activities of three antioxidant enzymes in the leaves, including catalase (CAT), peroxidase (POD), and superoxide dismutase (SOD). A catalase activity assay kit (BC0200, Jiancheng, Nanjing, China), peroxidase activity assay kit (BC0090, Jiancheng, Nanjing, China), and superoxide dismutase activity kit (BC0107, Jiancheng, Nanjing, China) were employed, as well as a malondialdehyde content kit (BC0020, Solar-bio, Beijing, China). Measurements were repeated three times for each treatment, and the choice of leaf material was the same as that in Section 4.3.

### 4.7. Analysis of Relationship

Correlation analysis of several key indicators and anthocyanins was conducted using R software (RStudio 4.2.2).

### 4.8. Detection of Gene Expression

This was carried out based on the method of Zhang et al. [64], with minor modifications. The second- and third-site leaves were removed from the plants at 0, 1, 3, 5, and 7 d. They were divided into three lyophilization tubes, quickly placed in liquid nitrogen for snap freezing, and stored at –80 °C in a refrigerator. Total RNA was extracted by referring to the instructions of FastPure Universal Plant Total RNA Isolation Kit (RC411, Vazyme Biotechnology Co., Ltd., Nanjing, China). cDNA was synthesized by referring to the instructions of the Evo M-MLV (Ekore, Hunan, China) reverse transcription kit. Primers were designed using the online primer design website (genscript). The primer sequences of each gene are shown in Table 2. The PCR reaction system was 2× SYBR Green Pro Taq HS Premix * 6 5 μL, 0.8 μL forward and 0.8 μL reverse primers, 1 μL cDNA, ddH_2_O to 10 μL. The PCR amplification conditions were: (1) 95 °C 30 s; and (2) 95 °C 5 s, 60 °C 30 s, 72 °C 10 min, 40 cycles, 65 °C 5 s, 95 °C 5 s, repeated three times. The relative expression of the target gene was calculated according to the 2−ΔΔCt method.

### 4.9. Statistical Analysis

Statistical analysis was performed using SPSS 19.0 software (SPSS Inc., Chicago, IL, USA), and differences were considered significant at *p* < 0.05. Tables were generated using Excel software (Microsoft Office Excel 2019; 18.2304.1202.0); and graphs were produced using SigmaPlot 12.5 software package (Systat Software GmbH, Erkrath, Germany), R software (RStudio, 4.2.2), and GraphPad Prism (8.4.3).

## 5. Conclusions

In summary, we studied the leaf color variation mechanism of *L. chinense* var. *rubrum* “Xiangnong Fendai” under WL, BL, BL + UL, and UL treatments at different times. The BL treatment was more conducive to the leaf color becoming red and an increase in anthocyanin accumulation. This was due to the positive response of receptor CRY to the BL signal, which upregulated the expression of HY5-BBX. It then induced MYB to promote the expression of AS structural genes, thereby increasing anthocyanin accumulation. The UL treatment promoted an increase in the chlorophyll and carotenoid contents in the leaves, and induced the upregulation of antioxidant enzyme activity and its related genes to better offset the damage caused by ROS to the plants. The results of the combined BL + UL light may be because the effect of UL on plants is much more significant than BL, so the plants under this treatment group were less favorable than those in the BL treatment group. The WL containing a full spectrum of light was flat and did not have a more significant positive or negative impact on the plants. Therefore, in the subsequent production, we may use BL to increase the ornamental and medicinal value of *L. chinense* var. *rubrum*.

## Figures and Tables

**Figure 1 plants-12-02169-f001:**
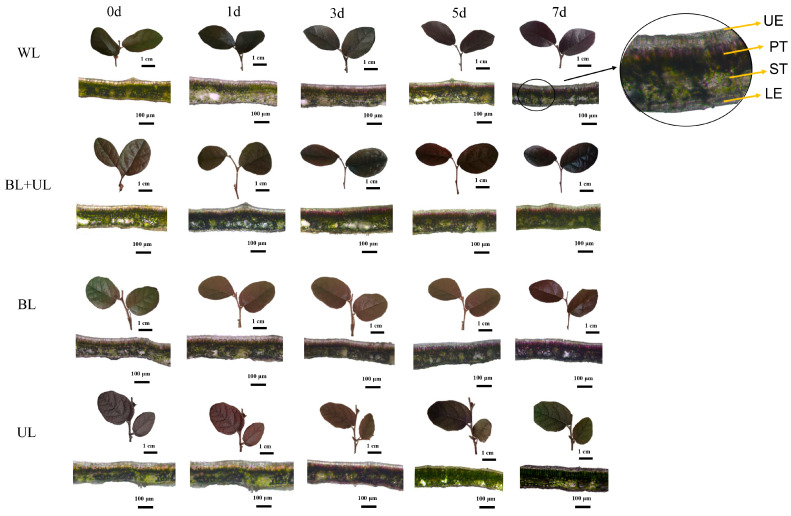
“Xiangnong Fendai” leaf color changes with time under white light (WL), blue light + ultraviolet-A light (BL + UL), blue light (BL), and ultraviolet-A light (UL) treatments. UE: upper epidermis; PT: palisade tissue; ST: spongy tissue; LE: lower epidermis.

**Figure 2 plants-12-02169-f002:**
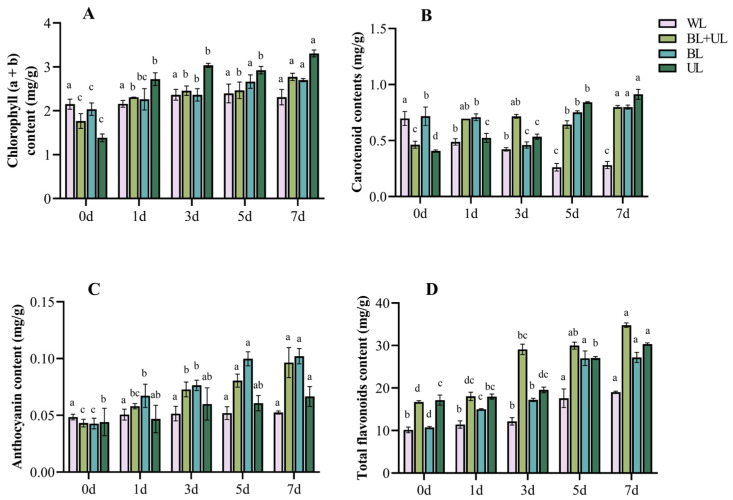
Changes in pigment content in leaves with time under different light quality treatments. White light (WL), blue light + ultraviolet-A light (BL + UL), blue light (BL), and ultraviolet-A light (UL). (**A**) Chlorophyll content (a + b); (**B**) carotenoid content; (**C**) anthocyanin content; and (**D**) total flavonoid content. The data in the figure are mean ± standard error; different lowercase letters are significantly different based on Tukey tests (*p* < 0.05).

**Figure 3 plants-12-02169-f003:**
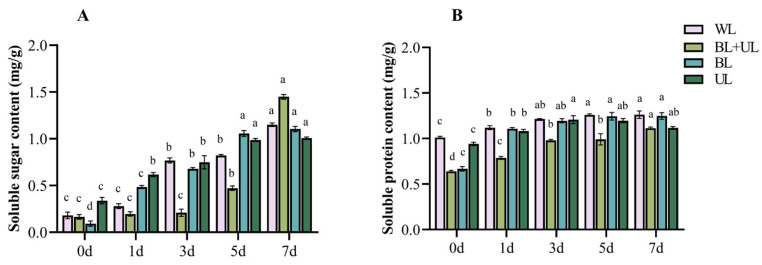
Changes in soluble sugar content (**A**) and soluble protein content (**B**) in leaves under different light quality treatments. White light (WL), blue light + ultraviolet-A light (BL + UL), blue light (BL), and ultraviolet-A light (UL). The data in the figure are mean ± standard error; different lowercase letters are significantly different based on Tukey tests (*p* < 0.05).

**Figure 4 plants-12-02169-f004:**
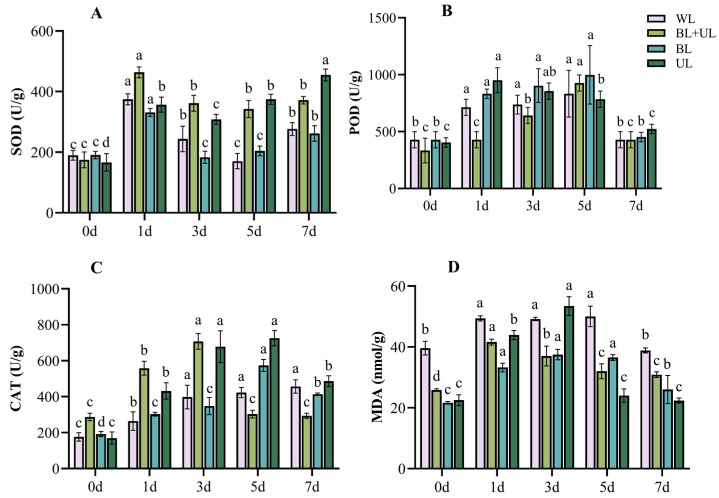
Changes in the activity of the antioxidant enzymes SOD (**A**), POD (**B**), and CAT (**C**), and in MDA content (**D**) in leaves with time under different light quality treatments. White light (WL), blue light + ultraviolet-A light (BL + UL), blue light (BL), and ultraviolet-A light (UL). The data in the figure are mean ± standard error; different lowercase letters are significantly different based on Tukey tests (*p* < 0.05).

**Figure 5 plants-12-02169-f005:**
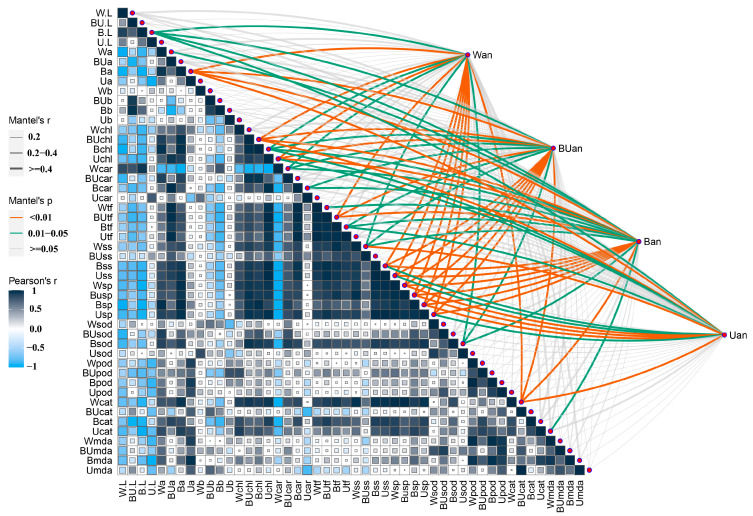
A pairwise comparison of anthocyanin content with other physicochemical factors under different light qualities is shown, with a color gradient denoting Pearson’s correlation coefficient.

**Figure 6 plants-12-02169-f006:**
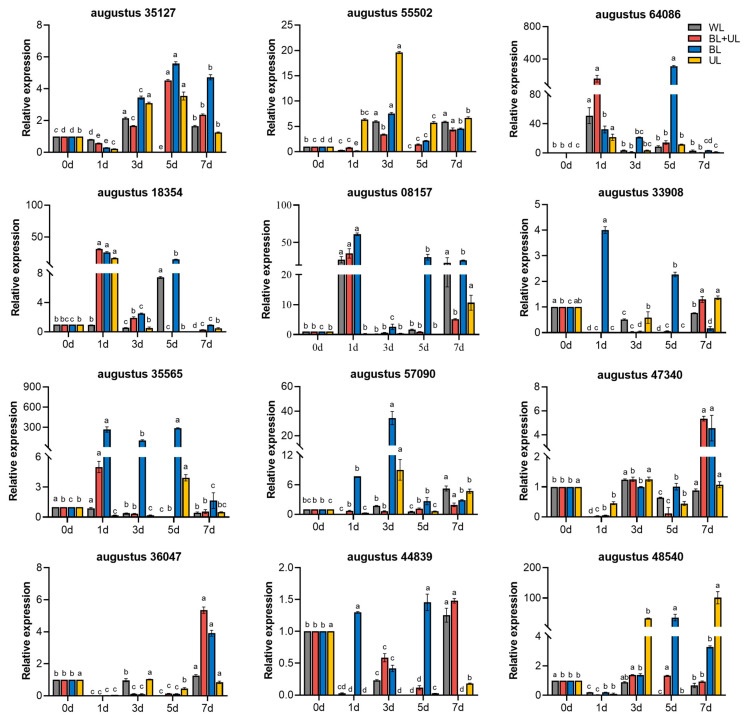
Changes in the relative expression of similar genes related to anthocyanin synthesis in the leaves of “Xiangnong Fendai” under different light quality treatments with time. White light (WL), blue light + ultraviolet-A light (BL + UL), blue light (BL), and ultraviolet-A light (UL). The data in the figure are mean ± standard error; different lowercase letters are significantly different based on Tukey tests (*p* < 0.05).

**Figure 7 plants-12-02169-f007:**
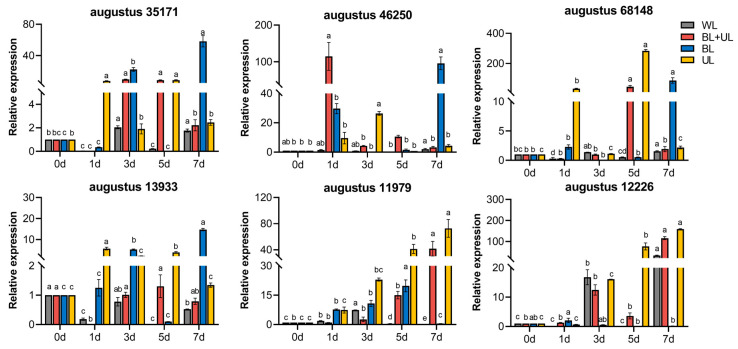
Changes in the relative expression of similar genes related to antioxidant enzyme synthesis in “Xiangnong Fendai” leaves under different light quality treatments with time. White light (WL), blue light + ultraviolet-A light (BL + UL), blue light (BL), and ultraviolet-A light (UL). The data in the figure are mean ± standard error; different lowercase letters are significantly different based on Tukey tests (*p* < 0.05).

**Figure 8 plants-12-02169-f008:**
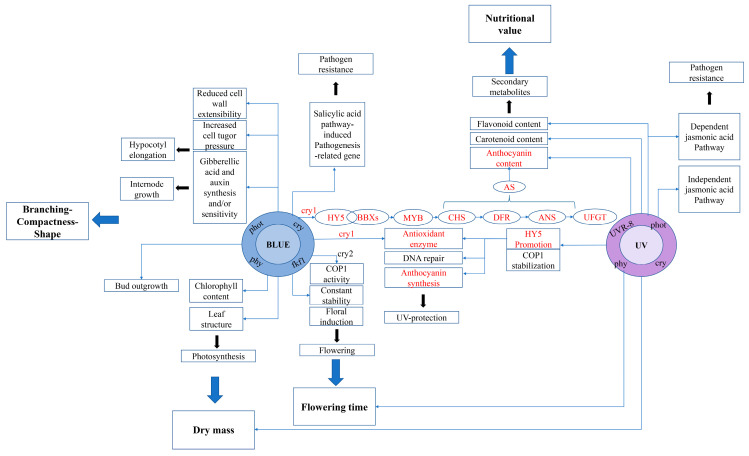
Regulatory network of the genes related to plant growth and development under blue and ultraviolet light (red font is the main direction of this experiment). Plant photoreceptors absorbing blue radiation: phot: phototropins; phy: phytochromes; fkf1: flavin-binding Kelch; cry: cryptochromes.

**Table 1 plants-12-02169-t001:** Determination of Munsell color system and color classification of “Xiangnong Fendai” leaves under light quality treatment.

Treatment Number	Treatment Time	Munsell Color System			Color Group According to CIELab
		H	V	C	
WL	0 d	2.4 GY	4.90	2.00	Olive green
	1 d	0.05 Y	3.90	2.00	Olive brown
	3 d	0.0034 Y	3.90	1.90	Olive brown
	5 d	7.1 R	3.10	1.50	Red
	7 d	9.9 YR	3.90	2.00	Yellowish-brown
BL + UL	0 d	1.2 GY	4.90	2.10	Olive green
	1 d	8.9 Y	4.70	2.60	Olive
	3 d	8.9 Y	4.70	2.60	Olive
	5 d	4.4 YR	3.10	0.94	Brown
	7 d	9.9 YR	3.90	1.90	Yellowish-brown
BL	0 d	2.5 GY	4.80	1.80	Olive green
	1 d	0.14 Y	3.90	2.00	Olive brown
	3 d	5.1 YR	3.40	2.00	Brown
	5 d	6.5 R	3.00	1.50	Red
	7 d	0.53 YR	3.50	1.80	Reddish-brown
UL	0 d	1.5 GY	4.80	2.10	Olive green
	1 d	2 Y	4.20	1.50	Olive brown
	3 d	7 YR	3.40	2.50	Brown
	5 d	1.2 Y	3.90	2.80	Olive brown
	7 d	1.5 GY	4.80	1.60	Olive green

Note: H, V, and C represent hue, value, and chroma, respectively.

**Table 2 plants-12-02169-t002:** Real-time fluorescence quantitative PCR primer sequence information of “Xiangnong Fendai”.

Gene Type	Gene Name		Primer Sequence (5′-3′)
Internal reference genes	*β-actin 2*	F	CCACAAGGCTTATTGATAGAAT
		R	CAATGGTTGAACCTGAATACT
*CRY*-like gene-1	*augustus 35127*	F	CTGGCGTCATCGATTCCATC
		R	AGTGGGTCTCTTCAGCAACA
*CRY*-like gene-2	*augustus 55502*	F	GCTGGCATGAGAGAGTTGTG
		R	TCTCCATGGAAGCTGCAGAA
*HY5*-like gene-1	*augustus 64086*	F	CAGCAAGCCCGAGAAAGAAA
		R	ACCTGGACCTTGATCGTGTT
*HY5*-like gene-2	*augustus 18354*	F	TTCCTCCCTCACTGCTCAAG
		R	AAACTCTGCCTCATCACCCA
*BBX*-like gene	*augustus 08157*	F	TGCAGATTCATTGCCGTCAG
		R	AACGTTTCTTGGGTGGCTTC
*MYB*-like gene	*augustus 33908*	F	CTACGCTTCTGCTGACGATG
		R	ATTGCAGGTTTCCGATGGTG
*CHS*-like gene-1	*augustus 35565*	F	AGTCGGTTCGGATCCACTAC
		R	CCTTCGCTATCCGGGAGAAT
*CHS*-like gene-2	*augustus 57090*	F	GACAGTGATGAAGCTCGCAA
		R	TTCTTCAGCACTCCTTCGGT
*DFR*-like gene-1	*augustus 47340*	F	GCCAACAATAGCTGGCATGT
		R	AGCCGGTCATCTTTACGCTA
*DFR*-like gene-2	*augustus 36047*	F	GCCAACAATAGCTGGCATGT
		R	AGCCGGTCATCTTTACGCTA
*ANS*-like gene	*augustus 44839*	F	AGTTGGAGGGATGGAAGAGC
		R	TTGGCCGTGATCCATTTGTC
*UFGT*-like gene	*augustus 48540*	F	GACGTCGTTCATGCTCCAAA
		R	CAACTCACCTTCCTCCCTGT
*SOD*-like gene-1	*augustus 35171*	F	GCCAAGGGAGATTCGTCAAC
		R	ACCACCTCCTTCATGGACAG
*SOD*-like gene-2	*augustus 13933*	F	ATCCTGCTGGGAAAGAGCAT
		R	CAACAACAGCCCTTCCAACA
*POD*-like gene-1	*augustus 46250*	F	TAGCCTCTCTTGCCACCAAA
		R	GTGGGACACACGTCGTAAAG
*POD*-like gene-2	*augustus 11979*	F	GTGGCCCTGAATACAACGTC
		R	TTTGGTGGCAAGAGAGGCTA
*CAT*-like gene-1	*augustus 68148*	F	GGCGTGAGAAGTGCGTTATT
		R	GACACGTGGGTCGGTTAAAG
*CAT*-like gene-2	*augustus 12226*	F	ATGAGGAGGCTGCAAGGATT
		R	AGGCTGCAAGGGAAAGAGAT

## Data Availability

Data are contained within the article.

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
