# Peer review of "Phenotypic, Physiological, and Molecular Response of Loropetalum chinense var. rubrum under Different Light Quality Treatments Based on Leaf Color Changes"

_plants, 2023, doi:10.3390/plants12112169_

Round 1

Reviewer 1 Report

In this manuscript, Zhang et al. did a classical comparative investigation to understand the effect of different light qualities on the leaf colour of an ornamentally important plant. Overall, the manuscript is well-written and well-presented. The results are adequately interpreted with relevant discussion. However, there are several comments listed below that need to be addressed, therefore, I would like to recommend this manuscript for a “Major revision”.

Comment 1: Line 61, Is it appropriate to use the term “light regulation”? Instead, I would prefer specific attributes of light here, for example, light intensity or light spectrum. I would also suggest using specific acronyms to denote different light spectrums such Blue light could be “BL” as used in material and method sections for better consistency.

Comment 2: Line 90, define “MDA” when used for the first time. The same also needs to be done for anthocyanin biosynthetic pathway genes

Comment 3: Lines 93-95, rewrite the sentences with more clarity.

Comment 4: Line 104, correct “SIHY5” to “SIHY5”. Use standard notions for writing gene and protein names. For example, SIHY5 will be SIHY5 (Gene), SIHY5 (Protein), and SIhy5 (mutant).

Comment 5: The quality of Figure 1 is very poor and needed to be improved.

Comment 6: Lines 425, I am wondering why we need an electron microscope for just observing the manually prepared cross sections. Kindly check and provide appropriate details.

Comment 7: Line 443-444, delete the repeated sentence.

Comment 8: Line 468-474, provide details for peroxidase (POD) activity assay.

Comments 9:

Some minor corrections/ comments

Comment 1: Lines 3, 95, 100, 107 correct the botanical name to Italics.

Comment 2: Line 39, delete repeated “found that”.

Comment 3: Line 102, correct “leu” to “Leu”.

Comment 4: Line 113, correct “Light” to “light”.

Comment 5: Line 118, write the scientific name of “ray florets” with proper reference.

The quality of the English language is largely adequate. The writing can still be improved in some places. I would suggest thoroughly reading the entire manuscript to avoid some minor mistakes. I tried to highlight some of such mistakes in my comments.

Author Response

Dear Reviewer,

Thank you for your detailed review of our manuscript entitled “Phenotypic, physiological and molecular response of Loropetalum chinense var. rubrum under different light quality treatments based on leaf color changes” (plants-2387260). The comments are of great help to improving the manuscript. We have studied the comments carefully and performed corresponding corrections in the revised manuscript. The point-by-point responses to the comments and suggestions are listed below.

“In this manuscript, Zhang et al. did a classical comparative investigation to understand the effect of different light qualities on the leaf colour of an ornamentally important plant. Overall, the manuscript is well-written and well-presented. The results are adequately interpreted with relevant discussion. However, there are several comments listed below that need to be addressed; therefore, I would like to recommend this manuscript for a “Major revision”.”

Comment 1: Line 61, Is it appropriate to use the term “light regulation”? Instead, I would prefer specific attributes of light here, for example, light intensity or light spectrum. I would also suggest using specific acronyms to denote different light spectrums such Blue Light could be “BL” as used in material and method sections for better consistency.

Response 1: Thank you for your suggestion. Your description is more accurate. According to your suggestion, we have replaced “light regulation” with “light intensity or spectrum” (Line 60).

In addition, we also accepted your great suggestion and unified the name of “Blue Light”. In addition to the first part of the abstract and the part of the introduction in “Materials and methods”, the rest uses the abbreviation “BL”.

Comment 2: Line 90, define “MDA” when used for the first time. The same also needs to be done for anthocyanin biosynthetic pathway genes.

Response 2: Thank you for your suggestion. We have included specific explanations and definitions of the abbreviations that first appeared. The additional details are as follows: Line 89-91, “Malondialdehyde (MDA) is one of the products of membrane lipid peroxidation, which indirectly reflects ROS damage to cells and organisms. Moreover, it was also used as one of the detection indexes of cell senescence when facing and enduring damage [21].” Line 117-120, “Li et al. [36] found that BL can increase the expression of the structural genes chalcone synthase (CHS) (the first key enzyme for the plant synthesis of flavonoids) and dihydroflavonol 4-reductase (DFR) (a key enzyme in the flavonoid biosynthesis pathway) in anthocyanin synthesis in Chrysanthemum (Chrysanthemum × morifolium Ramat.). There are also many results showing that the expression differences of most AS genes, such as CHS, DFR, anthocyanidin synthase (ANS) (a key enzyme at the end of the plant anthocyanin biosynthesis pathway, catalysing the conversion of colorless asssssssssss into colored anthocyanins), and UDP-glucose: flavonoid 3-O-glucosyltransferase (UFGT) (the last enzyme in the process of anthocyanin synthesis, which can catalyse unstable anthocyanins into anthocyanins)…”.

Comment 3: Lines 93-95, rewrite the sentences with more clarity.

Response 3: Thank you for your suggestion. We rewrite this sentence in Lines 92-96, “Antioxidant enzymes and anthocyanins play an essential role in the antioxidant process and show a complex relationship. Light-induced increased anthocyanins synthesis and antioxidant enzyme activity of Quercus [22].” as “Antioxidant enzymes and anthocyanins play a crucial role in plant antioxidant processes, and they have a complex relationship. Among them, it has been reported that anthocyanin content and antioxidant enzyme activity were proportionally increased after light induction in Quercus [22].”

Comment 4: Line 104, correct “SIHY5” to “SIHY5”. Use standard notions for writing gene and protein names. For example, SIHY5 will be SIHY5 (Gene), SIHY5 (Protein), and SIhy5 (mutant).

Response 4: Thank you for your suggestion. In our manuscript, we correct “SIHY5” to “SIHY5” (Line 106). In the future, we will pay more attention to the professional writing between genes, proteins and mutants.

Comment 5: The quality of Figure 1 is very poor and needed to be improved.

Response 5: Thank you for your suggestion. We re-adjust the size and resolution of Figure 1 to facilitate reading. Line 174, we've re-inserted high-quality pictures into the manuscript.

Figure 1. Leaves color changes of 'Xiangnong Fendai' with time under WL, BL+UL, BL, and UL treatments, UE: upper epidermis; PT: palisade tissue; ST: spongy tissue; LE: lower epidermis.

Comment 6: Lines 425, I am wondering why we need an electron microscope for just observing the manually prepared cross sections. Kindly check and provide appropriate details.

Response 6: Thank you for your suggestion. After our examination, it was found that the microscope we used was an optical microscope rather than an electron microscope. In Lines 466-469, we have corrected it in our manuscript, “After cutting several strips with a razor blade, a thin enough cross-section of the leaf was selected, covered with a coverslip, and the distribution of pigments in the cross-section structure was observed under an optical microscope (Leica D-35578, Wetzlar, Germany) with the photographs saved.”. Thank you for your earnestness to make us correct this mistake in time.

Comment 7: Line 443-444, delete the repeated sentence.

Response 7: Thank you for your suggestion. We deleted the repetitive sentences in our manuscript (Line 483).

Comment 8: Line 468-474, provide details for peroxidase (POD) activity assay.

Response 8: Thank you for your suggestion. We have supplemented the details for peroxidase (POD) activity assay in line 514, “Peroxidase activity assay kit (BC0090, Jiancheng, Nanjing, China),”.

Comments 9: Some minor corrections/ comments

Comment 9-1: Lines 3, 95, 100, 107 correct the botanical name to Italics.

Response 9-1: Thank you for your suggestion. We have replaced Lines 3, 95, 100, and 107 (the botanical name) with italics. Line 2-3: “Loropetalum chinense var. rubrum” change to “Loropetalum chinense var. rubrum”; Line 95-96: “Quercus” change to “Quercus”; Lines 100: “Solanum lycopersicum” change to “Solanum lycopersicum”; Lines 109: “Solanum melongena” change to “Solanum melongena”.

Comment 9-2: Line 39, delete repeated “found that”.

Response 9-2: Thank you for your suggestion. In line 39, we have deleted the repeated: “found that”.

Comment 9-3: Line 102, correct “leu” to “Leu”.

Response 9-3: Thank you for your suggestion. In line 103, we have corrected “leu” to “Leu”.

Comment 9-4: Line 113, correct “Light” to “light”.

Response 9-4: Thank you for your suggestion. In line 114, we have corrected “Light” to “light”.

Comment 9-5: Line 118, write the scientific name of “ray florets” with proper reference.

Response 9-5: Thank you for your suggestion. Line 121, we have corrected “ray florets” to the correct scientific name “Chrysanthemum (Chrysanthemum × morifolium Ramat.)” in our manuscript.

Once again, thank you very much for your comments and suggestions. A revised manuscript is attached. Should you have any questions, please contact us without any hesitation.

Sincerely yours,

Yanlin Li

14th May, 2023

Reviewer 2 Report

The aim of the research was to evaluate the influence of light of different colours on the colour of leaves. Physiological and biochemical metabolism (the content of pigments such as carotenoids and xanthophylls) was assessed. The content of anthocyanins and flavonoids was also tested. The work has been written correctly. The reasoning and discussion were done properly. Correct statistical methods were used to develop the research results.

However, the work should be supplemented with a research hypothesis. After minor corrections, the result can be published in Plants

Author Response

Dear Reviewer,

Thank you for your detailed review of our manuscript entitled “Phenotypic, physiological and molecular response of Loropetalum chinense var. rubrum under different light quality treatments based on leaf color changes” (plants-2387260). The comments are of great help to improving the manuscript. We have studied the comments carefully and perform corresponding corrections in the revised manuscript. The point-by-point responses to the comments and suggestions are listed below.

“The aim of the research was to evaluate the influence of light of different colours on the colour of leaves. Physiological and biochemical metabolism (the content of pigments such as carotenoids and xanthophylls) was assessed. The content of anthocyanins and flavonoids was also tested. The work has been written correctly. The reasoning and discussion were done properly. Correct statistical methods were used to develop the research results.

However, the work should be supplemented with a research hypothesis. After minor corrections, the result can be published in Plants”

Comment 1: The work should be supplemented with a research hypothesis.

Response 1: Thank you for your suggestion. Based on your suggestion, we added the research hypothesis in the introduction (Lines 136-151) to make the study more complete. The details are as follows, “According to the above introduction, we hypothesised that the reason for the red leaf color of L. chinense var. rubrum under light quality treatment might be that light affects the expression of anthocyanin-related genes to promote their synthesis and that the synthesis of anthocyanin can also actively increase resistance, and thus reduce the stress response of plants. This study used physiological, biochemical, and molecular verification methods to explore the light quality that can best promote color improvement in L. chinense var. rubrum leaves without causing light stress and to verify the relationship between some physical and chemical indicators and anthocyanin content under different light qualities. By analyzing the changes in pigment, soluble sugar and protein, antioxidant enzyme activity, and anthocyanin synthesis–related genes in leaves under BL and UL, the mechanism of color improvement in L. chinense var. rubrum leaves was examined. An effective L. chinense var. rubrum leaf color improvement scheme was proposed, which improved its market competitiveness and provided a theoretical basis for artificially enriching the leaf color of L. chinense var. rubrum. Finally, in our research discussions, we hope to translate this basic scientific knowledge into opportunities for agricultural production and to provide ideas for future research directions on the subject.”.

Once again, thank you very much for your comments and suggestions. A revised manuscript is attached. Should you have any questions, please contact us without any hesitation.

Sincerely yours,

Yanlin Li

14th May, 2023

Reviewer 3 Report

This is an interesting paper, and I appreciate the tremendous efforts the authors put in this work.  The authors have tracked morphological, biochemical and some transcriptomic changes in leaves of Loropetalum chinense var. rubrum, a well-known ornamental plant, in response to different light quality. These observations and data will be of certain interest for plant biologists and floriculturists in both science and business. There are, however, a few significant shortcomings in the way the paper is written and presented.

1.     The whole paper will certainly benefit from been checked by a professional editor: many sentences are unclear, e.g.

Line 65 (leaf color is “relatively single”);

Line 120 (“There are many factors have shown that.. under blue and ultraviolet light can activate..” – not clear);

Line 411, last sentence;

Line 422 – how one can make observations with bare hands?..

etc..

There are also unnecessary repetitious of the same information (e.g. Line 146 (respectively); lines 441-444).

2.     Materials and methods need improvement.

Part 4.2. – almost completely unclear (what was the use of LED lamps? What was the distance between plants? Were these plants in the climate control room during the experiment or in a greenhouse?)

Part 4.3. The first paragraph is not clear. In the second paragraph, what kind of a colorimetric instrument was used (please provide the model and the manufacturer)?

Part 4.4. What were the criteria for leaf selection? If the authors selected random leaves from the same branch, I would expect high variation in all parameters within one treatment. Was this the case?

Parts  4.5 and 4.6. Same question: what leaves were selected? Based on what criteria? Not clear.

3.     Please explain all abbreviations when they appear in the text for the first time.

4.     Unfortunately, most figures provided with the submission are very small. It was difficult to see the color change in Fig 1 or legends in Figs 4 and 5, Fig. 8 was almost unreadable.

5.     A note: The authors suggest that light quality may be used to improve the trade quality of plant material. But such color change would be transient, right? As a customer, my interest would be what happens to leaf color when I plant the material under ambient light conditions after one month or one season?

Extensive English editing is recommended. A final polish by a professional scientific editor will be a plus.

Author Response

Dear Reviewer,

Thank you for your detailed review of our manuscript entitled “Phenotypic, physiological and molecular response of Loropetalum chinense var. rubrum under different light quality treatments based on leaf color changes” (plants-2387260). The comments are of great help to improving the manuscript. We have studied the comments carefully and perform corresponding corrections in the revised manuscript. The point-by-point responses to the comments and suggestions are listed below.

“This is an interesting paper, and I appreciate the tremendous efforts the authors put in this work. The authors have tracked morphological, biochemical and some transcriptomic changes in leaves of Loropetalum chinense var. rubrum, a well-known ornamental plant, in response to different light quality. These observations and data will be of certain interest for plant biologists and floriculturists in both science and business. There are, however, a few significant shortcomings in the way the paper is written and presented.”

Comment 1: The whole paper will certainly benefit from been checked by a professional editor: many sentences are unclear, e.g.

1.1 Line 65 (leaf color is “relatively single”);

1.2 Line 120 (“There are many factors have shown that. under blue and ultraviolet light can activate.” – not clear);

1.3 Line 411, last sentence;

1.4 Line 422 – how one can make observations with bare hands?

1.5 There are also unnecessary repetitious of the same information (e.g. Line 146 (respectively);

1.6 lines 441-444).

Response 1: Thank you for your suggestion. According to your suggestion, we polished the manuscript and modified the following parts:

1.1 Line 65 (leaf color is “relatively single”);

Response 1.1: Lines 65-66 (leaf color is “relatively single”) have been changed to “The leaf color control method is relatively simple, so it is imperative to increase the convenience of the manual intervention leaf color method.”;

1.2 Line 120 (“There are many factors have shown that. under blue and ultraviolet light can activate.” – not clear);

Response 1.2: Line 120 (“There are many factors have shown that under blue and ultraviolet light can activate.” – not clear) has been changed to “A number of studies have shown that the expression of MYB can be activated under BL and UV light in apples [39-41], purple pakchoi [42], tomato [43-44], peach [45], and other peels to varying degrees, thereby regulating the up-regulation of anthocyanin synthesis–related genes, and ultimately promoting red-colored peel results.” (Line 128-131);

1.3 Line 411, last sentence;

Response 1.3: Line 411, last sentence (“Maintain all cultivation substrate ratio for garden soil: vermiculite: perlite = 3: 1: 1, consistent water and fertilizer management.”) has been changed to “Water and fertilizer management that maintained the same ratio of cultivation substrate was used for all experimental materials, with the following soil conditions: garden soil: vermiculite: perlite = 3:1:1.” (Line 443-445).

1.4 Line 422 – how one can make observations with bare hands?

Response 1.4: Line 422 – how one can make observations with bare hands? (“Observation of cross-sectioning with bare hands”). For a clearer description, the sentence was also rewritten as (Line 459-469) “The microscopic observation of the pigment distribution in the transverse structure of the leaf was carried out as follows. The second and third young leaves in a branch, counted from the top of the branch (approximately 5-6 leaves), were selected for observation under different treatments and at different time points (the selection criteria for the following test leaves were the same to ensure the consistency of the conditions). Distilled water was added to the middle of a clean slide and potatoes were cut into strips to be used as fixed objects, with a small opening at the front of the strip to hold the leaf in the middle. After cutting several strips with a razor blade, a thin enough cross-section of the leaf was selected, covered with a coverslip, and the distribution of pigments in the cross-section structure was observed under an optical microscope (Leica D-35578, Wetzlar, Germany) with the photographs saved.”.

1.5 There are also unnecessary repetitious of the same information (e.g. Line 146 (respectively);

Response 1.5: Line 146 (respectively): Line 157, the duplicate word has been removed;

1.6 lines 441-444).

Response 1.6: Lines 441-444: Duplicate sentences “The leaves were weighed with the veins removed, weighed 0.1 g, cut into 0.1 cm strips and treated with 95% ethanol for 24 h at 4 °C in the dark until the leaves turned white.” have been deleted (Line 483-484).

Comment 2: Materials and methods need improvement.

2.1 Part 4.2. almost completely unclear (what was the use of LED lamps? What was the distance between plants? Were these plants in the climate control room during the experiment or in a greenhouse?)

2.2 Part 4.3. The first paragraph is not clear. In the second paragraph, what kind of a colorimetric instrument was used (please provide the model and the manufacturer)?

2.3 Part 4.4. What were the criteria for leaf selection? If the authors selected random leaves from the same branch, I would expect high variation in all parameters within one treatment. Was this the case?

2.4 Parts 4.5 and 4.6. Same question: what leaves were selected? Based on what criteria? Not clear.

Response 2: Thank you for your suggestion. According to your suggestion, we modified the following parts:

2.1 Part 4.2. almost completely unclear (what was the use of LED lamps? What was the distance between plants? Were these plants in the climate control room during the experiment or in a greenhouse?)

Response 2.1 Part 4.2.: We have supplemented and modified this part in line 447-456. “Based on previous studies, light sources more favorable for anthocyanin synthesis were utilized as our experimental light sources, so as to screen for light sources more favorable for anthocyanin synthesis in L. chinense var. rubrum. In order to remove other influencing factors, the following experiments were carried out in the artificial climate control room with custom-made LED lamps: white light (WL) (plant growth spectrum), WL as control, blue light plus UV-A (BL+UL) (460 nm + 320 nm) (light ratio 1:1), blue light (BL) (460 ± 5 nm), and UV-A (UL) (320–400 nm). The light intensity was controlled by a light intensity meter at approximately 200 μmol/m2/s. The artificial climate control room had a light cycle precisely regulated by an automatic control switch (day/night: 14 h/10 h), a temperature of 24 °C/20 °C (day/night), and a humidity of 65-75%.”

2.2 Part 4.3. The first paragraph is not clear. In the second paragraph, what kind of a colorimetric instrument was used (please provide the model and the manufacturer)?

Response 2.2 Part 4.3.: We rewrite the unclear first paragraph in lines 461-471 to make it more readable, “Observation of leaf transverse structures by freehand sectioning: The second and third young leaf in branch, counted from the top of the branch (about 5-6 leaves), were selected for observation under different treatments and at different time points. Distilled water was added to the middle of a clean slide and potatoes were cut into strips to be used as fixed objects, with a small opening at the front of the strip to hold the leaf in the middle. After cutting several strips with a razor blade, a thin enough cross-section of the leaf was selected, covered with a coverslip and the distribution of pigments in the cross-section structure was observed under an optical microscope (Leica D-35578, Wetzlar, Germany) and the photographs were saved.”)

In the second paragraph we add the model number of the colorimetric instrument and the manufacturer in our manuscript (Line 473-474): “All the measured values of L*, a*, and b* were quantitatively derived using the YS 3020 spectrophotometer (3nh, China).”.

2.3 Part 4.4. What were the criteria for leaf selection? If the authors selected random leaves from the same branch, I would expect high variation in all parameters within one treatment. Was this the case?

Response 2.3 Part 4.4: In order to maintain a relatively good consistency in the selection of materials and reduce errors as much as possible, we did not randomly select leaves in a branch, but unified the leaf position of the branch. We all selected the second and third leaves for observation and determination. At the same time, we supplemented all parts of the process in the manuscript to make it clearer. Such as Part 4.3 (462-465) “The second and third young leaves in a branch, counted from the top of the branch (approximately 5-6 leaves), were selected for observation under different treatments and at different time points (the selection criteria for the following test leaves were the same to ensure the consistency of the conditions)”.

2.4 Parts 4.5 and 4.6. Same question: what leaves were selected? Based on what criteria? Not clear.

Response 2.4 Parts 4.5 and 4.6: We also supplemented the selection of leaves in these parts to make these parts more clearly described in Lines 507-511 and Lines 514-522, respectively:

“4.5. Determination of total flavonoids, soluble sugar, and soluble protein content

The contents of total flavonoids, soluble sugar, and soluble protein in the leaves were determined using kits (total flavonoids assay kit: ZCSO873, Shanghai zcibio technology Co., Ltd., China; plant soluble sugar assay kit: BC0030, Solar-bio, Beijing, China; plant protein quantification assay kit: A045-2, Jiancheng, Nanjing, China), and the choice of leaf material was the same as that in Part 4.3.” (Line 507-511).

4.6. Determination of MDA content and antioxidant enzyme activity

We investigated the malondialdehyde (MDA) content and the activities of three antioxidant enzymes in the leaves, including catalase (CAT), peroxidase (POD), and superoxide dismutase (SOD). A catalase activity assay kit (BC0200, Jiancheng, Nanjing, China), peroxidase activity assay kit (BC0090, Jiancheng, Nanjing, China), and superoxide dismutase activity kit (BC0107, Jiancheng, Nanjing, China) were employed, as well as a malondialdehyde content kit (BC0020, Solar-bio, Beijing, China). Measurements were repeated three times for each treatment, and the choice of leaf material was the same as that in Part 4.3.” (Line 514-522).

Comment 3: Please explain all abbreviations when they appear in the text for the first time.

Response 3: Thank you for your suggestion. According to your suggestion, we have added specific explanations and definitions of the abbreviations that first appeared. The additional details as follows:

Line 89-91, “Malondialdehyde (MDA) is one of the products of membrane lipid peroxidation, which indirectly reflects ROS damage to cells and organisms. Moreover, it was also used as one of the detection indexes of cell senescence when facing and enduring damage [21].”

Line 80-82, “Light quality caused the accumulation of unwanted and harmful reactive oxygen species (ROS) and various physiological impairments, which affected the metabolism of anthocyanins.”;

Line 104-105 “directly targets constitutively photomorphogenic 1 (COP1) and acts as a central positive regulator of light signaling.”; Line 107-109, “MYB1” modified to “MYB1(v-myb avian myeloblastosis viral oncogene homolog) binding to downstream anthocyanin structural (AS) genes, further inducing anthocyanin accumulation in Solanum melongena.”;

Line 118-128, “Li et al. [36] found that BL can increase the expression of structural genes Chalcone synthase (CHS) (the first key enzyme for plant synthesis of flavonoids) and dihydroflavonol 4-reductase (DFR) (a key enzyme in flavonoid biosynthesis pathway) in anthocyanin synthesis in Chrysanthemum (Chrysanthemum × morifolium Ramat.). There are also many results showings that the expression differences of most AS structural genes, such as CHS, DFR, anthocyanidin synthase (ANS) (It is a key enzyme at the end of the plant anthocyanin biosynthesis pathway, catalyzing the conversion of colorless anthocyanins into colored anthocyanins.) and UDP-glucose: flavonoid 3-O-glucosyltransferase (UFGT) (It is the last enzyme in the process of anthocyanin synthesis, which can catalyze unstable anthocyanins into anthocyanins.), depend on the expression of transcription factors (such as MYB) [37-38]”.

Comment 4: Unfortunately, most figures provided with the submission are very small. It was difficult to see the color change in Fig 1 or legends in Figs 4 and 5, Fig. 8 was almost unreadable.

Response 4: Thank you for your suggestion. We have modified these figures’ sizes and resolutions in our revised edition to facilitate reading; at the same time, other graphs are also checked. We adjusted the quality of Figure 1 as much as possible to make it more straightforward, so we improved the size and resolution of Figure 1. For Figure 4 and Figure 5, we increased the size of the legend font to make it more transparent and readable; in Figure 8, we increased the font size to make it more straightforward.

Figure 1. Leaves color changes of 'Xiangnong Fendai' with time under WL, BL+UL, BL, and UL treatments, UE: upper epidermis; PT: palisade tissue; ST: spongy tissue; LE: lower epidermis.

Figure 4. Changes of three antioxidant enzyme activity and MDA content in leaves with time under different light quality treatments. Changes of SOD(A), POD(B), CAT(C), and MDA(D) with time under different light qualities.

Figure 5. A pairwise comparison of anthocyanin content with other physicochemical factors under different light qualities is shown, with a color gradient denoting Pearson's correlation coefficient.

Figure 8. Regulatory network of genes related to plant growth and development under blue and ultraviolet light (Red font is the main direction of this experiment). Plant photoreceptors absorbing blue radiations: phot, phototropins; phy, phytochromes; fkf1, Flavin-binding Kelch; cry, crypto-chromes.

Comment 5: A note: The authors suggest that light quality may be used to improve the trade quality of plant material. But such color change would be transient, right? As a customer, my interest would be what happens to leaf color when I plant the material under ambient light conditions after one month or one season?

Response 5: Thank you very much for your interesting questions. In our experiment, we mainly studied the physiological and biochemical indexes of WL, BL, BL + UL and UL at 0d, 1d, 3d, 5d and 7d, and the expression of key genes and antioxidant enzyme genes in anthocyanin synthesis in different treatments. Through the correlation analysis of physiological and biochemical indexes, gene expression differences and anthocyanin content of leaves before and after treatment, the response mechanism of different light quality treatments in a relatively short time was revealed. It was also found that under the short-term treatment of BL, it was more conducive to the reddening of the leaves of L. chinense var. rubrum and did not lead to excessive photooxidation. Thank you very much for your interesting questions again, but we only conducted this experiment in a short period of time, and have not yet conducted the experiment in the environment you mentioned, so we have not yet reached an effective conclusion, but we are also very interested in your question. In future research, we will try to observe what happens to leaf when I plant the material under ambient light conditions after one month or one season.

Comments on the Quality of English Language: Extensive English editing is recommended. A final polish by a professional scientific editor will be a plus.

Response: Thank you for your suggestion. We have professionally polished the manuscript. The certificate of polishing is as follows:

Once again, thank you very much for your comments and suggestions. A revised manuscript is attached. Should you have any questions, please contact us without any hesitation.

Sincerely yours,

Yanlin Li

14th May, 2023

Reviewer 4 Report

This manuscript studies the responses of Loropetalum chinense leaves color under four different light regimes e.g. white light, blue light, ultraviolet-A light and blue plus ultraviolet-A light. The duration of light treatments was 7 days and the measurements were taken every second day. Pigment (chlorophyll, carotenoid, anthocyanin) concentrations, soluble sugars and proteins as well as antioxidant enzymes were studied. In addition, gene expression related to anthocyanin synthesis was detected.

Many recent studies have been done in this field, showing that the light quality induces changes to the plant physiology and metabolism and results in improved of plant growth.

The paper is well written and structured and adds new information on the activation of genes related to anthocyanin synthesis expression.

The paper can be accepted by “Plants” after minor revision.

Specific comments:

Keywords: I think that the term “genes expression” should be added instead of physiological and biochemical indicators present in the Title.

Results: 2.1. Line 141. Table 1 not Table 2. More description of the cross section of the leaves under the light regimes is needed.

2.2. Color of WL shown in Fig 1 and Table 1 in the 5th day is red. This is not match with Fig 2C (anthocyanin content).

2.3. Comment about WL should be included.

2.4. MDA content reached its maximum value in 3rd day of UL and WL (lines 221-222). To be checked.

Materials and Methods: If the experiment was carried out under greenhouse or open air conditions, it should be referred.

Author Response

Dear Reviewer,

Thank you for your detailed review of our manuscript entitled “Phenotypic, physiological and molecular response of Loropetalum chinense var. rubrum under different light quality treatments based on leaf color changes” (plants-2387260). The comments are of great help to improving the manuscript. We have studied the comments carefully and perform corresponding corrections in the revised manuscript. The point-by-point responses to the comments and suggestions are listed below.

“This manuscript studies the responses of Loropetalum chinense leaves color under four different light regimes e.g. white light, blue light, ultraviolet-A light and blue plus ultraviolet-A light. The duration of light treatments was 7 days and the measurements were taken every second day. Pigment (chlorophyll, carotenoid, anthocyanin) concentrations, soluble sugars and proteins as well as antioxidant enzymes were studied. In addition, gene expression related to anthocyanin synthesis was detected.

Many recent studies have been done in this field, showing that the light quality induces changes to the plant physiology and metabolism and results in improved of plant growth.

The paper is well written and structured and adds new information on the activation of genes related to anthocyanin synthesis expression. The paper can be accepted by ‘Plants’ after minor revision.”

Comment 1: Keywords: I think that the term “genes expression” should be added instead of physiological and biochemical indicators present in the Title.

Response 1: Thank you for your suggestion. We have replaced the “physiological and biochemical indicators” of the Keywords with “genes expression” in our manuscript (Line 46).

Comment 2: Results: 2.1. Line 141. Table 1 not Table 2. More description of the cross section of the leaves under the light regimes is needed.

Response 2: Thank you for your suggestion. We have changed “Table 1” to “Table 2” (Line 154). In addition, we describe more about the cross-section of leaves under light conditions in Lines 167-173: “Furthermore, by observing the distribution of pigments in leaf cross-sections by freehand sectioning, we found that anthocyanins were mainly concentrated in the palisade tissue (PT) and less in the upper epidermis (UE), the lower epidermis (LE), and spongy tissue (ST), which largely determined the color we observed in the leaves. In the leaves under BL treatment, we could observe more clearly that the degree of redness in the palisade tissue was higher on 7 d than under WL, BL+UL, and UL treatments. The changes in pigmentation in the cross-sectional structure were largely consistent with the description of the changes in leaf color.”

Comment 3: Results: 2.2. Color of WL shown in Fig 1 and Table 1 in the 5th day is red. This is not match with Fig 2C (anthocyanin content).

Response 3: Thank you for your suggestion. Our previous study identified 185 flavonoids (including 15 anthocyanins, seven isoflavones, 104 flavones, 20 flavanones and 39 flavonols) and 22 polyphenols (including 19 phenolic acids and three proanthocyanidins) in L. chinense var. rubrum. The 13 anthocyanins, which belong to glycosides of peonidin, delphinidin, cyanidin, malvidin and petunidin, are the main phytochemicals of leaf pigments in L. chinense var. rubrum. Non-anthocyanin polyphenols include flavonoid glycosides, flavonol glycosides, flavanone glycosides, isoflavone glycosides, phenolic acids, and proanthocyanidins are auxiliary pigments that affect the color of leaves of L. chinense var. rubrum. The significant difference in the content of these polyphenols in the leaves resulted in different colors of the leaves of different varieties of L. chinense var. rubrum [1]. Therefore, the leaf phenotype was red on 5d of WL treatment. Still, the anthocyanin content did not increase significantly, which may be due to the increase of some auxiliary pigments in the leaves, since some auxiliary pigments, dihydroflavonols are precursors for the synthesis of flavonols, which are synthesized under the action of FLS, and can be reduced to colorless anthocyanin aglycones under the act of DFR. FLS enzymes mainly affect leaf color in two ways: through auxiliary coloring, and through substrate competition. FLS and DFR competition for common dihydroflavonol substrates can change the ratio of flavonols to anthocyanins, thus affecting the final coloration of plants [2]. It can also be seen from the data of our total flavonoid content that the total flavonoid content increased significantly on the 5d of WL treatment compared with 0d, which means the proportion of other auxiliary pigments in leaves increases.

In addition, the content and proportion of various types of pigments in plant species, the pH value of the medium, metal atoms, and the internal and surface physical structures of the plants carrying pigments can directly or indirectly change people's perception of plant color [3].

Thank you again for this valuable suggestion, but this is only one of our speculations that has not been verified. We will try to verify this statement in future experiments. To avoid the same doubts from other readers in the future, we also added such an explanation in the discussion section (Part 3.2, Line 356-366) of the manuscript to explain this question: “From the results, we also found that the leaf phenotype under WL treatment at 5 d was significantly redder than that at 0 d, but the anthocyanin content did not increase significantly. From previous studies conducted by our group [51] identified 185 flavonoids and 22 polyphenols other than flavonoids in L. chinense var. rubrum (including 19 phenolic acids and three proanthocyanidins). Non-anthocyanin polyphenols such as flavonoid glycosides and phenolic acids are auxiliary pigment components that affect the leaf color of L. chinense var. rubrum. This phenomenon may be due to the increase in the content of some auxiliary pigments in the leaves. Our total flavonoid content data shows that the total flavonoid content was significantly increased at 5 d compared with 0 d. However, this is only one of our speculations and has not been verified, so we will try to substantiate this statement in future experiments.”.

Comment 4: Results: 2.3. Comment about WL should be included.

Response 4: Thank you for your suggestion. We have added more descriptions of WL in this paragraph. The whole paragraph has been rewritten as follows (Line 204-217): “As can be seen from Figure 3, the soluble sugar content and soluble protein content increased gradually over time under all light qualities, with the soluble sugar content increased significantly to the highest level at 7 d under BL treatment, increasing by 1087.35% compared with 0 d, followed by BL+UL treatment, increasing to 778.54% at 7 d. Compared with 0 d, the content increased by 538.89% under WL treatment, while under UL treatment, it increased the least, increasing by only 197.93% at 7 d compared with 0 d. The soluble protein content peaked at 3 d under UL treatment, with an increase of 28.72% over 0 d. At 7 d, the content under BL treatment peaked, increasing by 86.57% compared with 0 d, followed by the content under BL+UL treatment, with an increase of 74.33% over 0 d, while WL treatment only increased the content by 24.77% over 0 d. In the leaves of “Xiangnong Fendai”, BL treatment was more beneficial to a significant accumulation of soluble sugars. In contrast, treatment with UL was more beneficial to a significant accumulation of soluble proteins, followed by BL+UL treatment. In contrast, the increase in the two osmotic adjustment substances under WL treatment was poor.

Comment 5: MDA content reached its maximum value in 3rd day of UL and WL (lines 221-222). To be checked.

Response 5: Thank you for your suggestion. We rewrite this sentence in the revision edition as “The MDA content of the plant leaves under UL treatment reached the maximum value (53.4 nmol/g) on the third day, which was significantly higher than that at 0 d (137.3%). The maximum content was attained on the fifth day under WL treatment (50.04 nmol/g). The antioxidant enzyme activity and MDA content increased the most under UL treatment, followed by treatment with BL+UL.” (Line 236-240).

Comment 6: Materials and Methods: If the experiment was carried out under greenhouse or open air conditions, it should be referred.

Response 6: Thank you for your suggestion. In this section, we have described the experiments, which were carried out in the artificial climate control room. The specific descriptions have been added to the manuscript in Lines 449-453, with the main processes as follows: “To remove other influencing factors, the following experiments were carried out in the artificial climate control room with custom-made LED lamps: white light (WL) (plant growth spectrum), WL as control, blue light plus UV-A (BL+UL) (460 nm + 320 nm) (light ratio 1:1), blue light (BL) (460 ± 5 nm), and UV-A (UL) (320-400 nm).”

Reference:

  1. Chen, Q. R.; Cai, W. Q.; Zhang, X.; Zhang, D. M.; LI, W. D.; Xu, L.; Yu, X. Y.; Li, Y. L., The Comparative Studies on Phyto-chemicals of Leaf Coloration of Loropetalum chinense rubrum. Acta Horticulturae Sinica x, 48 (10), 1969-1982.
  2. Vu TT, Jeong CY, Nguyen HN, et al. Characterization of Brassica napus flavonol synthase involved in flavonol biosynthesis in Brassica napus L. J. Agric. Food Chem, 2015, 63: 7819-7829.
  3. Wu Q., Li P.C., Zhang H.J., et al. Relationship between the flavonoid composition and flower colour variation in Victoria[J]. Plant Biology, 2018, 20:674-68.

Once again, thank you very much for your comments and suggestions. A revised manuscript is attached. Should you have any questions, please contact us without any hesitation.

Sincerely yours,

Yanlin Li

14th May, 2023

Round 2

Reviewer 1 Report

na

Author Response

Dear reviewer,

    Thank you for your help!

    Best wishes!

    Yanlin Li,

    23th May, 2023

Reviewer 3 Report

The quality of the manuscript was greatly improved from the original version. Congratulations. Still, I suggest some minor, mainly linguistic, edits to improve the smooth flow of the text. Please take into account that I am not a native English speaker and you can made different edits by your choice, but please pay attention to the lines/sentences I mention below – they need improvement or clarification.  See the attached file for details.

Some minor edits are suggested in the attached file.

Author Response

Dear Reviewer,

Thank you for your detailed review of our manuscript entitled “Phenotypic, physiological and molecular response of Loropetalum chinense var. rubrum under different light quality treatments based on leaf color changes” (plants-2387260). Thank you very much for your careful inspection and professional advice. We have studied the comments carefully and perform corresponding corrections in the revised manuscript. The point-by-point responses to the comments and suggestions are listed below.

“The quality of the manuscript was greatly improved from the original version. Congratulations. Still, I suggest some minor, mainly linguistic, edits to improve the smooth flow of the text. Please take into account that I am not a native English speaker and you can made different edits by your choice, but please pay attention to the lines/sentences I mention below – they need improvement or clarification.  See the attached file for details.”

Comment 1: Abstract. “Light quality is a vital environmental signal used to trigger growth and to develop structural differentiation in plants, and it controls morphological, physiological, and biochemical metabolites. In previous studies, anthocyanins were regulated by different light qualities.” Light cannot control metabolites (not directly). The second sentence is too vague. May I suggest that the author who wrote the last paragraph of the Introduction edit this part?

Response 1: Thank you for your suggestion. According to your suggestion, we have rewritten the “controls” to “influences” to make it more accurate. (Line 29) In addition, we rewrite the second sentence as: “In previous studies, different light qualities can regulate the synthesis of anthocyanin.” (Line 30)

Comment 2: Line 38. “antioxidant enzyme activity “ – please mention the enzymes

Response 2: Thank you for your suggestion. We have changed “antioxidant enzyme activity” into “the activities of three antioxidant enzymes in the leaves, including catalase (CAT), peroxidase (POD), and superoxide dismutase (SOD), to varying degrees over time.” (Line 38-39).

Comment 3: Line 51. Remove “essentially”

Response 3: Thank you for your suggestion. We have removed “essentially” from this sentence: “is a woody plant with characteristic vividly colorful leaves and flowers that is easily shaped and trimmed….” (Line 56).

Comment 4: Line 53. “other bioactive components”

Response 4: Thank you for your suggestion. We have replaced “other components”  with “other bioactive components”. (Line 58)

Comment 5: Line 54. “, e.g. the leaves can be..”

Response 5: Thank you for your suggestion. We have rewritten as “, e.g. the leaves can be..”.(Line 59)

Comment 6: Line 56. “..has become an important plant for gardening [4],..”

Response 6: Thank you for your suggestion. We have replaced “has become an important product of garden production [4],” with “has become an important plant for gardening [4],..” in our manuscript. (Line 61)

Comment 7: Line 57. Change “components” to “composition”

Response 7: Thank you for your suggestion. We have changed “components” to “composition” in our manuscript. (Line 62)  

Comment 8: Line 60. “water supply”

Response 8: Thank you for your suggestion. We have changed “water, and light intensity or light spectrum” to “water supply, and light intensity or light spectrum” in our manuscript. (Line 64)

Comment 9: Lines 63-63. “In the current cultivation of L. chinense var. rubrum, due to the flat leaf color and vulnerability to adverse environmental impacts, market competitiveness is low.” –Consider changing to “Varying leaf color and vulnerability to environmental stresses reduce the market value of this plant”

Response 9: Thank you for your suggestion. We have changed “In the current cultivation of L. chinense var. rubrum, due to the flat leaf color and vulnerability to adverse environmental impacts, market competitiveness is low.” to “Varying leaf color and vulnerability to environmental stresses reduce the market value of this plant” in our manuscript. (Line 68-69)

Comment 10: Line 65. “The leaf color control method is relatively simple, so it is imperative to increase the convenience of the manual intervention leaf color method.” – maybe change to “Hence, it is desirable to develop simple and convenient method to control the leaf color for improving the market value of the plant”. Would that be correct?

Response 10: Thank you for your suggestion. We agree that the modified language description is more appropriate, so we modified this sentence to “Hence, it is desirable to develop simple and convenient method to control the leaf color for improving the market value of the plant” in our manuscript. (Line 69-71)

Comment 11: Line 67. “with light quality treatment” – change to “by varying light quality”

Response 11: Thank you for your suggestion. We have changed “with light quality treatment” to “by varying light quality” in our manuscript. (Line 71)

Comment 12: Line 71. Remove “of plants”

Response 12: Thank you for your suggestion. We have removed “of plants” in our manuscript. (Line 75)

Comment 13: Line 73. “Likewise, the anthocyanins were accumulated in response to intense illumination stimulation, which was accompanied by increasing the content of soluble sugars and soluble proteins.” – this is what is meant in the sentence?

Response 13: Thank you for your suggestion. In order to clarify the meaning of this sentence, we have rewritten it as: “Similarly, with the increase of soluble sugar and soluble protein under BL and UV-A, it also leads to the accumulation of anthocyanins.” (Line 77-78)

Comment 14: Line 75. “These primary metabolites were the precursors of the flavonoid metabolism.” –Actually, a precursor is phenylalanine and a number of other compounds, not sugars, not proteins. I suggest you to remove this whole sentence.

Response 14: Thank you for your suggestion. We have removed this sentence: “These primary metabolites were the precursors of the flavonoid metabolism.” in our manuscript. (Line 78)

Comment 15: Line 76. “Furthermore, soluble sugar significantly promoted the carbon metabolism of Liquidambar formosana, making its leaves redder and darker under BL exposure [16]” –Wrong reference? There is no such information in the publication cited. In addition, the statement is very strange to me.

Response 15: Thank you for your suggestion. We have rewritten this sentence: “The soluble sugar content under blue light treatment was the highest, which effectively promoted the carbon metabolism of Liquidambar formosana leaves, thus providing raw materials for the large synthesis of anthocyanins, thus promoting the redness of leaves.” in our manuscript. (Line 79-82). We also checked the references and re-insert the correct references in the manuscript. (Line 760-761)

Comment 16: Line 79. “deepen” – in what way? Maybe replace by “caused changes in leaf color”?

Response 16: Thank you for your suggestion. We have changed “deepen” to “caused changes in leaf color” in our manuscript. (Line 83)

Comment 17: Line 80. “Light quality caused the accumulation of unwanted and harmful reactive oxygen species (ROS) and various physiological impairments, which affected the metabolism of anthocyanins.” – consider changing to “Light of specific wavelength may cause the accumulation of unwanted and harmful reactive oxygen species (ROS) in plant leaves, also affecting the metabolism of anthocyanins.”

Response 17: Thank you for your suggestion. We have changed “Light quality caused the accumulation of unwanted and harmful reactive oxygen species (ROS) and various physiological impairments, which affected the metabolism of anthocyanins.” to “Light of specific wavelength may cause the accumulation of unwanted and harmful reactive oxygen species (ROS) in plant leaves, also affecting the metabolism of anthocyanins.” in our manuscript. (Line 85-87)

Comment 18: Line 87. “in the body” – “in plants”?

Response 18: Thank you for your suggestion. We have changed “in the body” to “in plants” in our manuscript. (Line 92)

Comment 19: Line 95. What means “light induction”?

Response 19: Thank you for your suggestion. “light induction” means that “light treatment”, we have replaced “light induction” with “light treatment” to make it more readable. (Line 100)

Comment 20: Line 100 “strain” – was it “mutant”? Also, change “of overexpression” to “with overexpression”.

Response 20: Thank you for your suggestion. “strain” does not mean mutant, but species. In case of misunderstanding, we have deleted this word in our manuscript. (Line 105). What’s more, we have changed “of overexpression” to “with overexpression”. (Line 105)

Comment 21: Line 110. Strains (mutants?) of what species?

Response 21: Thank you for your suggestion. We have revised and supplemented this sentence in our manuscript: “SIHY5 silencing and CRY1a overexpression led to a significant decrease in anthocyanin accumulation in Arabidopsis thaliana (L.) Heynh and other plants.” (Line 162-163)

Comment 22: Line 111. “of BL triggered” – did you mean “triggered by BL”?

Response 22: Thank you for your suggestion. Yes, that means “triggered by BL”, and we have changed “of BL triggered” to “triggered by BL” in our manuscript. (Line 164)

Comment 23: Line 143. “also actively increase resistance”.. to what? Maybe remove this part of the sentence?

Response 23: Thank you for your suggestion. We have deleted this sentence in the manuscript to make it smoother. (Line 195)

Comment 24: Line 160. Please define “H”, “V” and “C”.

Response 24: Thank you for your suggestion. We add the definition: “The Munsell color system (Hue, Value, and Chroma; It will be abbreviated as H, V, C later.)” in our manuscript. (Line 220)

Comment 25: Table 1. Please define H, C and V in the footnote.

Response 25: Thank you for your suggestion. We have added footnotes to the table to define “H”, “V”, and “C”. (Line 246)

Comment 26: Line 187. “As shown in Figure 2, chlorophyll (a+b) content increased with the time of light quality treatment, and the increase in pigment content was not apparent under WL treatment. However, the chlorophyll (a+b) content increased significantly after BL+UL, BL, and UL treatments, reaching the maximum at 7 d. ” – Consider changing to “As shown in Figure 2, chlorophyll (a+b) content increased over time under different light qualities reaching a maximum at day 7, except for WL where the increase was insignificant.”

Response 26: Thank you for your suggestion. We have changed “As shown in Figure 2, chlorophyll (a+b) content increased with the time of light quality treatment, and the increase in pigment content was not apparent under WL treatment. However, the chlorophyll (a+b) content increased significantly after BL+UL, BL, and UL treatments, reaching the maximum at 7 d.” to “As shown in Figure 2, chlorophyll (a+b) content increased over time under different light qualities reaching a maximum at day 7, except for WL where the increase was insignificant.” (Line 249-251)

Comment 27: Line 222. “In contrast, the increase in the two osmotic adjustment substances under WL treatment was poor.” – I suggest deleting this sentence.

Response 27: Thank you for your suggestion. According to your suggestion, we have deleted this sentence in our manuscript. (Line 291)

Comment 28: Figure 3. Legend. Consider modifying to “Changes in soluble sugar content (A) and soluble protein content (B) in leaves under different light quality treatments. The data in the figure are mean ± standard error; different lowercase letters are significantly different based on Tukey tests (p < 0.05)”

Response 28: Thank you for your suggestion. According to your suggestion, we have changed “Changes in two osmotic adjustment substances in leaves with time under different light quality treatments. A: changes in soluble sugar content of “Xiangnong Fendai” leaves with time under different light qualities; B: changes in soluble protein content of “Xiangnong Fendai” leaves with time under different light qualities. The data in the figure are mean ± standard error; different lowercase letters are significantly different based on Tukey tests (p < 0.05).” to “Changes in soluble sugar content (A) and soluble protein content (B) in leaves under different light quality treatments. The data in the figure are mean ± standard error; different lowercase letters are significantly different based on Tukey tests (p < 0.05)” in our manuscript. (Line 297-300)

Comment 29: Line 233. “It can be seen from Figure 4 that, under UL conditions, the activities of three antioxidant enzymes were basically higher.” – Consider changing to “It can be seen from Figure 4 that, under UL, the activities of antioxidant enzymes CAT, SOD and POD were generally higher compared to other light treatments.”

Response 29: Thank you for your suggestion. According to your suggestion, we have changed “It can be seen from Figure 4 that, under UL conditions, the activities of three antioxidant enzymes were basically higher.” to “It can be seen from Figure 4 that, under UL, the activities of antioxidant enzymes CAT, SOD and POD were generally higher compared to other light treatments.” in our manuscript. (Line 303-304)

Comment 30: Line 235. “In addition, WL (374.5 U/g), BL + UL (463.5 U/g), and BL (331.0 U/g) treatments all caused the plant leaves to produce extensive enzyme activity at 1 d, and then decrease” –Consider changing to “The activity of SOD in leaves peaked at day one of exposure to WL (374.5 U/g), BL + UL (463.5 U/g), and BL (331.0 U/g), and then decrease”

Response 30: Thank you for your suggestion. According to your suggestion, we have changed “In addition, WL (374.5 U/g), BL + UL (463.5 U/g), and BL (331.0 U/g) treatments all caused the plant leaves to produce extensive enzyme activity at 1 d, and then decrease” to “The activity of SOD in leaves peaked at day one of exposure to WL (374.5 U/g), BL + UL (463.5 U/g), and BL (331.0 U/g), and then decrease” in our manuscript. (Line 306-307)

Comment 31: Line 237. “POD and CAT enzymes increased first, and then decreased under various light qualities.” – This does not correctly describe the figures. Maybe remove the sentence?

Response 31: Thank you for your suggestion. According to your suggestion, we have removed this sentence in our manuscript. (Line 307)

Comment 32: Line 240. “…while WL (832.2 U/g), BL + UL (927.3 U/g) and BL (998.6U/g) treatments reached the maximum at 5 d” – Consider changing to “…while under WL (832.2 U/g), BL + UL (927.3 U/g) and BL (998.6U/g) treatments it reached the maximum at day 5”.

Response 32: Thank you for your suggestion. According to your suggestion, we have changed “…while WL (832.2 U/g), BL + UL (927.3 U/g) and BL (998.6U/g) treatments reached the maximum at 5 d” to “…while under WL (832.2 U/g), BL + UL (927.3 U/g) and BL (998.6U/g) treatments it reached the maximum at day 5” in our manuscript. (Line 309-310)

Comment 33: Line 247. Consider changing to “Therefore, the antioxidant enzyme activity and MDA content increased the most under UL treatment, followed by treatment with BL+UL.”

Response 33: Thank you for your suggestion. According to your suggestion, we have changed “The antioxidant enzyme activity and MDA content increased the most under UL treatment, followed by treatment with BL+UL.” to “Therefore, the antioxidant enzyme activity and MDA content increased the most under UL treatment, followed by treatment with BL+UL.” in our manuscript. (Line 316-317)

Comment 34: Figure 4. Legend. Consider changing to “Changes in the activity of the antioxidant enzymes SOD (A), POD (B), and CAT (C), and in MDA content (D) in leaves with time under different light quality treatments. The data in the figure are mean ± standard error; different lowercase letters are significantly different based on Tukey tests (p < 0.05).”

Response 34: Thank you for your suggestion. According to your suggestion, we have changed “Changes in the activity of three antioxidant enzymes and MDA content in leaves with time under different light quality treatments. Changes in SOD(A), POD(B), CAT(C), and MDA(D) with time under different light qualities. The data in the figure are mean ± standard error; different lowercase letters are significantly different based on Tukey tests (p < 0.05).” to “Changes in the activity of the antioxidant enzymes SOD (A), POD (B), and CAT (C), and in MDA content (D) in leaves with time under different light quality treatments. The data in the figure are mean ± standard error; different lowercase letters are significantly different based on Tukey tests (p < 0.05).” in Figure 4. Legend. (Line 319-323)

Comment 35: Line 257. “Under BL+UL treatment, anthocyanin content and chlorophyll content in leaves, there was a significant positive correlation between total flavonoid content and soluble protein content.” - Consider changing to “Under BL+UL treatment, there was a significant positive correlation between total flavonoid content and soluble protein content.”

Response 35: Thank you for your suggestion. According to your suggestion, we have changed “Under BL+UL treatment, anthocyanin content and chlorophyll content in leaves, there was a significant positive correlation between total flavonoid content and soluble protein content.” to “Under BL+UL treatment, there was a significant positive correlation between total flavonoid content and soluble protein content.” in our manuscript. (Line 327-362)

Comment 36: Line 274. Under BL treatment, at the relative expression of augustus 35127 and augustus 55502 was 5.59 and 7.55-fold higher at 5d and 3d than at 0d; augustus 55502 reached its maximum at 3d under WL, BL and UL treatments, significantly higher than at 0d, and 5.99-fold, 7.54-fold and 277 19.61-fold higher than at 0d, respectively. “ – This sentence is confusing and difficult to understand. Maybe split in two? For example, “Under BL treatment, at the relative expressions of augustus 35127 and augustus 55502 were 5.59 and 7.55-fold higher at 5d and 3d than at 0d. Augustus 55502 expression reached its maximum at 3d under WL, BL and UL treatments, which was 5.99-fold, 7.54-fold and 19.61-fold higher than at 0d, respectively.”

Response 36: Thank you for your suggestion. According to your suggestion, we have changed “Under BL treatment, at the relative expression of augustus 35127 and augustus 55502 was 5.59 and 7.55-fold higher at 5d and 3d than at 0d; augustus 55502 reached its maximum at 3d under WL, BL and UL treatments, significantly higher than at 0d, and 5.99-fold, 7.54-fold and 277 19.61-fold higher than at 0d, respectively.” to “Under BL treatment, at the relative expressions of augustus 35127 and augustus 55502 were 5.59 and 7.55-fold higher at 5d and 3d than at 0d. Augustus 55502 expression reached its maximum at 3d under WL, BL and UL treatments, which was 5.99-fold, 7.54-fold and 19.61-fold higher than at 0d, respectively.” in our manuscript. (Line 377-381)

Comment 37: Line 313. Also consider splitting the sentence in two: “The augustus 44839 gene reached maximum expression at 5 d under BL treatment, being 1.46 times higher than at 0 d. Its expression at 7 d under WL and BL+UL treatments was significantly (1.26 times and 1.48 times, respectively) higher than at 0 d”.

Response 37: Thank you for your suggestion. According to your suggestion, we have changed “The augustus 44839 gene reached maximum expression at 5 d under BL treatment, being 1.46 times higher than at 0 d, while the expressions under WL and BL+UL treatments were significantly higher at 7 d than at 0 d, being 1.26 and 1.48 times higher than at 0 d.” to “The augustus 44839 gene reached maximum expression at 5 d under BL treatment, being 1.46 times higher than at 0 d. Its expression at 7 d under WL and BL+UL treatments was significantly (1.26 times and 1.48 times, respectively) higher than at 0 d” in our manuscript. (Line 419-422)

Comment 38: Line 319.”gene” – “genes”

Response 38: Thank you for your suggestion. According to your suggestion, we have changed “gene” to “genes” in our manuscript. (Line 425)

Comment 39: In Figs 2-4 and 6,7 – please define light qualities in the legends as you did for Fig. 1: White light (WL), Blue light + ultraviolet-A light (BL+UL), Blue light (BL), and ultraviolet-A light (UL)

Response 39: Thank you for your suggestion. According to your suggestion, we have added this sentence, “White light (WL), Blue light + ultraviolet-A light (BL+UL), Blue light (BL), and ultraviolet-A light (UL)” in Figures 2-4, and in Figures 6-7. (Line 274, 298, 320, 429, 463)

Comment 39: Line 363. “scientific” – maybe “fact-based” or “measurable”?

Response 39: Thank you for your suggestion. According to your suggestion, we have changed “scientific” to “fact-based” in our manuscript. (Line 478)

Comment 40: Line 582. “.. at different times”?

Response 40: Thank you for your suggestion. According to your suggestion, we have changed “a different times” to “at different times” in our manuscript. (Line 699)

Comment 41: Line 679. Volume 427?

Response 41: Thank you for your suggestion. According to your suggestion, we have changed “42 7” to “42 (7)” in our manuscript. (Line 802)

Comment 42: Line 761. Pages?

Response 42: Thank you for your suggestion. According to your suggestion, we have added the pages: “Zhang, D.; Chen, Q.; Zhang, X.; Lin, L.; Cai, M.; Cai, W.; Liu, Y.; Xiang, L.-S.; Sun, M.; Yu, X.; Li, Y., Effects of low temperature on flowering and the expression of related genes in Loropetalum chinense var. rubrum. Frontiers in Plant Science 2022, 13, 1000160.” in our manuscript. (Line 885)

Reviewer 4 Report

After incorporating my suggestions the masnuscript has significantly improved. It can be accepted in its present form.

Author Response

(The authors gave the same response as above.)
